# Ciliary Motility Decreased by a CO_2_/HCO_3_^−^-Free Solution in Ciliated Human Nasal Epithelial Cells Having a pH Elevated by Carbonic Anhydrase IV

**DOI:** 10.3390/ijms25169069

**Published:** 2024-08-21

**Authors:** Shota Okamoto, Makoto Yasuda, Kotoku Kawaguchi, Kasane Yasuoka, Yumi Kikukawa, Shinji Asano, Taisei Tsujii, Sana Inoue, Kikuko Amagase, Taka-aki Inui, Shigeru Hirano, Toshio Inui, Yoshinori Marunaka, Takashi Nakahari

**Affiliations:** 1Research Laboratory for Epithelial Physiology, Research Organization of Science and Technology, Ritsumeikan University BKC, Kusatsu 525-8577, Japan; k-kawagu@dent.meikai.ac.jp (K.K.); kasane.yasuoka@gmail.com (K.Y.); coconut2110@gmail.com (Y.K.); ashinji@ph.ritsumei.ac.jp (S.A.); ph0161vh@ed.ritsumei.ac.jp (T.T.); marunaka@koto.kpu-m.ac.jp (Y.M.); 2Department of Otolaryngology-Head and Neck Surgery, Graduate School of Medical Science, Kyoto Prefectural University of Medicine, Kyoto 602-8566, Japan; myasu@koto.kpu-m.ac.jp (M.Y.); inui1227@koto.kpu-m.ac.jp (T.-a.I.); hirano@koto.kpu-m.ac.jp (S.H.); 3Laboratory of Molecular Physiology, Faculty of Pharmacy, Ritsumeikan University BKC, Kusatsu 525-8577, Japan; 4Laboratory of Pharmacology and Pharmacotherapeutics, Faculty of Pharmacy, Ritsumeikan University BKC, Kusatsu 525-8577, Japan; sinoue@fc.ritsumei.ac.jp (S.I.); amagase@fc.ritsumei.ac.jp (K.A.); 5Saisei Mirai Clinics, 3-34-8 Okubocho, Moriguchi 570-0012, Japan; t-inui@saisei-mirai.or.jp; 6Medical Research Institute, Kyoto Industrial Health Association, Kyoto 604-8472, Japan

**Keywords:** nasal cilia, intracellular pH, NH_4_^+^ pulse, NBC, bicarbonate transport metabolon, airway, ALI, ciliary motility, ciliary waveform, CBF

## Abstract

An application of CO_2_/HCO_3_^−^-free solution (Zero-CO_2_) did not increase intracellular pH (pH_i_) in ciliated human nasal epithelial cells (c-hNECs), leading to no increase in frequency (CBF) or amplitude (CBA) of the ciliary beating. This study demonstrated that the pH_i_ of c-hNECs expressing carbonic anhydrase IV (CAIV) is high (7.64), while the pH_i_ of ciliated human bronchial epithelial cells (c-hBECs) expressing no CAIV is low (7.10). An extremely high pH_i_ of c-hNECs caused pH_i_, CBF and CBA to decrease upon Zero-CO_2_ application, while a low pH_i_ of c-hBECs caused them to increase. An extremely high pH_i_ was generated by a high rate of HCO_3_^−^ influx via interactions between CAIV and Na^+^/HCO_3_^−^ cotransport (NBC) in c-hNECs. An NBC inhibitor (S0859) decreased pH_i_, CBF and CBA and increased CBF and CBA in c-hNECs upon Zero-CO_2_ application. In conclusion, the interactions of CAIV and NBC maximize HCO_3_^−^ influx to increase pH_i_ in c-hNECs. This novel mechanism causes pH_i_ to decrease, leading to no increase in CBF and CBA in c-hNECs upon Zero-CO_2_ application, and appears to play a crucial role in maintaining pH_i_, CBF and CBA in c-hNECs periodically exposed to air (0.04% CO_2_) with respiration.

## 1. Introduction

Intracellular pH (pH_i_) regulates various cellular functions, including airway ciliary beating [1,2]. In many cell types, including the airway ciliated epithelial cell, the pH_i_ is controlled by CO_2_ and ion transporters, such as Na^+^/HCO_3_^−^ cotransporter (NBC), Cl^−^/HCO_3_^−^ exchanger (anion exchanger, AE) and Na^+^/H^+^ exchanger (NHE). The pH_i_ is controlled by the reaction mediated by carbonic anhydrase (CA) (Equation (1)).
CO_2_ + H_2_O ↔ H^+^ + HCO_3_^−^(1)

CA is an enzyme that catalyzes the hydration and dehydration of CO_2_. In general, experimental procedures, such as the switch to a CO_2_/HCO_3_^−^-free solution (Zero-CO_2_) from a CO_2_/HCO_3_^−^-containing solution (control solution) and activation of NBC (HCO_3_^−^ entry), shift Equation (1) to the left (an increase in pH_i_), and contrarily, a procedure such as the switch to the control solution from the Zero-CO_2_ shifts Equation (1) to the right (a decrease in pH_i_).

The ciliated nasal epithelium is a unique tissue placed under a low temperature [3] and an unusual CO_2_ condition. The CO_2_ concentration of the apical surface is changed from 0.04% (fresh air) to 1% (exhaled air) with non-exercised respiration. The periodic air exposure (0.04% CO_2_) appears to increase pH_i_, leading to a ciliary beat frequency (CBF) increase in the ciliated nasal epithelial cells, since the application of the Zero-CO_2_ has been shown to increase pH_i_, CBF and ciliary bend distance (CBD, an index of amplitude) in tracheal and lung airway ciliated cells [1,4,5]. To keep an adequate ciliary beating during the periodic exposure to the air (an extremely low CO_2_ concentration), the ciliated nasal epithelial cells need a special mechanism. For example, to maintain CBF when cells are exposed to a low temperature, ciliated nasal epithelial cells express the thermosensitive transient receptor potentials (TRP) A1 and M8 [3]. However, it still remains uncertain how ciliated nasal epithelial cells keep an adequate ciliary beating during a periodical exposure to low and high CO_2_ concentrations. Ciliated human nasal epithelial cells (c-hNECs) were differentiated by the air liquid interface (ALI) culture. In c-hNECs, an application of Zero-CO_2_ induced small transient increases in pH_i_, CBF and CBD [2], while it induced their large sustained increases in the tracheal airway ciliary cells [1]. Respectively, an application of an NH_4_^+^ pulse induced a gradual decrease in pH_i_ and CBF (acetazolamide (an inhibitor of CA)-sensitive) following an immediate increase in c-hNECs [2] but a sustained increase in tracheal airway ciliary cells [1]. The gradual decreases in CBF following immediate increases indicate decreases in pH_i_ in c-hNECs upon applications of the Zero-CO_2_ and the NH_4_^+^ pulse. Since these decreases were acetazolamide-sensitive [2], the application of the Zero-CO_2_ or the NH_4_^+^ pulse appears to shift Equation (1) to the right to increase H^+^ in c-hNECs, but not in tracheal ciliary cells. However, it remains uncertain what decreases pH_i_ (the right shift of Equation (1)) upon applying Zero-CO_2_ or the NH_4_^+^ pulse.

A previous study demonstrated that eleven CA subtypes are expressed in nasal epithelia, including CAIV (a membrane-associated CA) [6]. The CAIV plays important roles in the regulation of pH_i_ in several cell types [7,8,9,10]. The physical and functional interactions between CAIV and NBC have been shown in co-transfected HEK239 cells with NBC1b and CAIV, and they maximize the rate of HCO_3_^−^ transport [7,8]. The C-terminal tail of CAIV is anchored in the outer surface of the plasma membrane, and a physical interaction between extracellular CAIV and NBC1 occurs via the fourth extracellular loop of NBC1 [7]. In the basolateral membrane of renal proximal tubules, CAIV, which colocalizes with NBC1, increases NBC1 activity [9]. The R14W mutation of CAIV, which has been detected in an autosomal dominant form of retinitis pigmentosa, impairs pH balances of photoreceptor cells by affecting HCO_3_^−^ influx [10]. These findings suggest that CAIV interacts with NBC to increase the rate of HCO_3_^−^ transport in the c-hNECs. Moreover, NBC and AE have been shown to express in nasal epithelia [11]. Thus, CAIV may interact with NBC, forming the bicarbonate transport metabolon in c-hNECs [7,8,12].

We hypothesized that the interactions between CAIV and NBC would maximize the rate of HCO_3_^−^ influx in c-hNECs, leading to a high pH_i_ [7,8,12]. The high pH_i_ (low [H^+^]_i_) may shift Equation (1) to the right to maintain CBF or CBD or cause their negligibly small increases, even upon Zero-CO_2_ application, and to induce their gradual decreases during the NH_4_^+^ pulse. We also used ciliated human bronchial epithelial cells (c-hBECs), which were differentiated from normal human bronchial epithelial cells (NHBE) by the ALI [13]. We found that c-hBECs, similar to c-hNECs, express CAs except CAIV, NBCs and AEs. The c-hBECs appear to be a good model of airway ciliated cells expressing no CAIV. The goal of this study is to clarify the CAIV-mediated mechanism, which suppresses increases in CBF, CBD and pH_i_ in c-hNECs upon applying Zero-CO_2._

## 2. Result

### 2.1. CAIV Expression in c-hNECs but Not in c-hBECs

Eleven CA subtypes (I, II, III, IV, Va, Vb, VI, VII, IX, XII and XIV) have been shown to be expressed in normal nasal mucosa [6]. In the preliminary experiments, we examined expressions of ten CA isozyme mRNAs with the exception of CAVa in c-hNECs and c-hBECs by reverse transcription-polymerase chain reaction (RT-PCR) (Appendix A). The primers used are shown in Table 1. The mRNAs of ten CA subtypes were expressed in c-hNECs. Interestingly, in contrast to other analyzed CAs, the mRNA of CAIV was not expressed in c-hBECs. Kim et al. also reported that mRNAs of CA II, IV and Vb were highly expressed in normal nasal mucosa [6]. Based on these findings, the mRNA expressions of five CA isozymes (I, II, III, IV and Vb) were examined by RT-PCR in c-hNECs (Figure 1A) and c-hBECs (Figure 1B). The mRNA of CAIV is expressed in c-hNECs but not in c-hBECs. The analysis of CAIV mRNA expressions by real-time PCR revealed that mRNAs were significantly higher in c-hNECs than in c-hBECs (Figure 1C).

The mRNA expression levels of CA isoforms were low in nasal polyp tissues compared with normal nasal mucosa [6]. In this study, we used two cell types of c-hNECs (uncinate process and nasal polyp) of chronic sinusitis (CS) patients. The mRNAs of five CA subtypes were similarly expressed in two cell types of c-hNECs.

The western blot (WB) analysis is consistent with real-time PCR data. We used two anti-CAIV antibodies (MAB2186 and AF2186, R&D System). In WB using MAB2186, multiple bands were detected in both cells, and the band suggesting CAIV was detected at 46 kDa in c-hNECs but not in c-hBECs (Appendix A). We also examined WB using AF2186. CAIV is known to have two N-glycosylation sites, the sialations of which increase the molecular mass by 11–45 kDa [14]. Although the specific band for CAIV has been shown to be approximately 33 kDa (datasheet of AF2186 supplied from R&D systems, Minneapolis, MN, USA), the previous studies demonstrated that the band for CAIV is 46 kDa in co-transfected HEK293 cells with cDNAs encoding AE2, CAII and CAIV [7,8]. Therefore, we treated the lysate with PNGaseF (a recombinant glycosidase, Roche, Basel, Switzerland) overnight, before starting the western blotting. In WB using AF2186, a single band detected was decreased to 40 kDa but was still larger than 33 kDa. Thus, the treatment with PNGaseF did not completely break the glycosylation sites. Another treatment may be required.

In c-hBECs, the CAIV band was faint. The densities of the CAIV band were normalized by those of GAPDH. The normalized value of CAIV protein expression was significantly higher in c-hNECs (*n* = 4) than in c-hBECs (*n* = 4) (*p* < 0.01) (Figure 2B).

Immunofluorescence examinations for CAIV were carried out in c-hNECs (Figure 3A1–A4) and c-hBECs (Figure 3B1–B4) using AF2186 (anti-CAIV antibody, Figure 3A1,B1) and ab179484 (anti-alpha-tubulin (AC-tubulin) antibody, a cilia marker) (Figure 3A2,B2). The double staining showed that cilia existing in the apical surface are positively stained for CAIV and AC-tubulin in c-hNECs (Figure 3A3). However, in c-hBECs, no immunofluorescence for CAIV was detected in cilia (Figure 3B1–B3). Thus, CAIV exists in the apical cilia of c-hNECs but not in those of c-hBECs.

### 2.2. Expression of NBC and AE mRNAs Detected by RT-PCR in c-hNECs and c-hBECs

The mRNA expressions of NBCs and AEs examined by RT-PCR were similar in c-hNECs (Figure 4A,C) and c-hBECs (Figure 4B,D). The expression of NBC and AE mRNAs has already been shown in human nasal epithelia [11,15,16] and mice bronchiolar epithelia [17]. The expression of NBC1, NBCe1, NBCe2, NBCn1, NDCBE1 and AE2 mRNAs was similar in c-hNECs from uncinate process and nasal polyp tissues of CS patients. The primers used for detecting NBC and AE are shown in Table 2.

### 2.3. CBF and CBD in c-hNECs and c-hBECs

In c-hNECs perfused with the control solution aerated with 95% O_2_ and 5% CO_2_ at 37 °C (control perfusion), CBF was 8.39 ± 2.44 Hz (*n* = 47), and CBD was 62.2 ± 13.8 µm (*n* = 20). No difference in CBFs was noted between c-hNECs from nasal polyp and uncinate process tissues of CS patients. In c-hBECs, CBF was 8.37 ± 1.04 Hz (*n* = 12), and CBD was 65.3 ± 8.0 µm (*n* = 12) during the control perfusion. Thus, there was no statistical difference in CBF or CBD between c-hNECs and c-hBECs.

Next, we applied a CO_2_/HCO_3_^−^-free solution in c-hNECs and c-hBECs for understanding the effects of an extremely low CO_2_ concentration on CBF and pH_i_.

#### 2.3.1. Ciliated hNECs

##### Effects of CO_2_/HCO_3_^−^-Free Solution (Zero-CO_2_) on CBF, CBD and pH_i_ in c-hNECs

Figure 5A shows changes in CBF and CBD ratios in c-hNECs upon applying Zero-CO_2_. The application of Zero-CO_2_ induced a small transient increase followed by a gradual decrease in the CBF ratio or CBD ratio. The ratios of CBF and CBD 1.5 min after applying the Zero-CO_2_ were 1.02 (*n* = 15) and 1.01 (*n* = 11, *p* < 0.05), and those 10 min after the application were 0.91 and 0.88 (*p* < 0.05), respectively. Thus, the application of Zero-CO_2_ decreased both CBF and CBD ratios in c-hNECs. Changes in the pH_i_ were measured by the fluorescence ratio of BCECF (F490/F440) (Figure 5B). The pH_i_ of c-hNECs in the control solution was 7.64 ± 0.19 (*n* = 9). The application of Zero-CO_2_ gradually decreased pH_i_. The pH_i_ at 1 min after the application was 7.55 (*n* = 9), and that at 5 min after the application was 7.24. This result indicates that the application of Zero-CO_2_ induces the right shift of Equation (1) (H^+^ production) or a decrease in the left shift (reduced H^+^ elimination) due to the low level of HCO_3_^−^. Inui et al. (2020) showed small transient increases in CBF, CBD and pH_i_ in c-hNECs [2]. They kept cells at 4 °C until the start of the experiments to reduce cellular damages by suppressing cellular metabolism. The small transient increases may be caused by a low temperature surrounding cells. However, in spite of the condition of keeping cells at 4 °C, their increases were negligibly small compared with those reported by airway ciliated cells [1,2,4].

Since changes in CBF and CBD ratios were similar upon applying Zero-CO_2_, we used the CBF ratio to assess the activities of ciliary beating.

##### Effects of NH_4_^+^ Pulse on CBF and pH_i_ in c-hNECs

An increase in pH_i_ is well known to enhance CBF and CBD in airway ciliary cells, including c-hNECs [1,2,4]. In ciliary cells of the trachea and lung airways, the application of Zero-CO_2_ induced a large increase in pH_i_ [1], but it induced a small increase in pH_i_ in c-hNECs [2]. To understand the effects of an elevation of pH_i_ on CBF in c-hNECs, we examined the effects of pH_i_ elevation on CBF, using the NH_4_^+^ pulse.

We applied the NH_4_^+^ pulse in the Zero-CO_2_. The switch to the Zero-CO_2_ did not increase the CBF. Then, the application of the NH_4_^+^ pulse immediately increased CBF followed by a gradual decrease. The ratios of CBF at 1 min and 5 min from the start of the NH_4_^+^ pulse were 1.09 and 1.03 (*n* = 6), respectively. Thus, the CBF ratio significantly decreased during the NH_4_^+^ pulse (*p* < 0.05), suggesting a pH_i_ decrease during the NH_4_^+^ pulse in the Zero-CO_2_. The cessation of the NH_4_^+^ pulse immediately decreased CBF (Figure 6A). Inui et al. demonstrated that the gradual CBF decrease during the NH_4_^+^ pulse is inhibited by acetazolamide (a nonspecific CA inhibitor) in c-hNECs [2]. These results suggest that Equation (1) shifts to the right (H^+^ production) during the NH_4_^+^ pulse in the Zero-CO_2_. A high value of pH_i_ and a low [HCO_3_^−^]_i_ appear to shift Equation (1) to the right in c-hNECs.

We also examined the effects of an NH_4_^+^ pulse on CBF and pH_i_ in the control solution (5% CO_2_). The application of the NH_4_^+^ pulse immediately increased CBF and pH_i_ and then gradually decreased them. The cessation of the NH_4_^+^ pulse immediately decreased CBF and pH_i_ (Figure 6B,C). The pH_i_ at 1 min from the start of the NH_4_^+^ pulse was 8.08, and that at 7 min was 7.65 (*n* = 9, *p* < 0.05), indicating that c-hNECs produce H^+^ in the control solution during application of the NH_4_^+^ pulse. A high pH_i_ appears to shift Equation (1) to the right during the NH_4_^+^ pulse.

##### Effects of CA Inhibitors on CBF and pH_i_ Changed by Applying the CO_2_/HCO_3_^−^-Free Solution and NH_4_^+^ Pulse in c-hNECs

The CA, especially CAIV, appears to be a key enzyme to regulate pH_i_ and CBF in c-hNECs. We first examined the effects of dorzolamide (1 µM, an inhibitor of CAII and CAIV) on CBF in c-hNECs. The addition of dorzolamide did not change the CBF ratio (0.98 at 5 min after the addition, *n* = 16), and the subsequent application of Zero-CO_2_ did not change the CBF ratio (0.99 at 5 min after the application) (Figure 7A). The effects of brinzolamide (0.1 μM, a CAII inhibitor) on CBF were examined. The addition of brinzolamide did not change the CBF ratio (1.00 at 5 min after the addition, *n* = 21), and the subsequent application of Zero-CO_2_ also did not (0.98 at 5 min after the application) (Figure 7B). The effects of brinzolamide on CBF were examined upon applying the NH_4_^+^ pulse. In the presence of brinzolamide, the NH_4_^+^ pulse induced an immediate increase followed by a gradual increase in CBF. The CBF ratio was 1.07 (*n* = 7) at 1 min and 1.17 at 5 min from the start of the NH_4_^+^ pulse. Cessation of the NH_4_^+^ pulse immediately decreased CBF (Figure 7C). In the presence of brinzolamide, the NH_4_^+^ pulse increased CBF without any decrease in c-hNECs, indicating that Equation (1) shifts to the right (H^+^ production) during the NH_4_^+^ pulse via CAII. We also measured pH_i_ during application of the NH_4_^+^ pulse. Brinzolamide alone did not affect the pH_i_ in c-hNECs. The pH_i_s were 7.81 (*n* = 6) before and 7.69 at 5 min after the addition of brinzolamide. The application of the NH_4_^+^ pulse increased and plateaued pH_i_, and the pH_i_s were 8.46 at 1 min (*n* = 6) and 8.41 at 5 min from the start of the NH_4_^+^ pulse. Cessation of the NH_4_^+^ pulse decreased pH_i_ (Figure 7D). The effects of dorzolamide and brinzolamide on CBF were similar upon applying the Zero-CO_2_ solution or the NH_4_^+^ pulse, although CAII is a cytosolic enzyme and CAIV is a membrane-associated enzyme attached by a glycosylphosphatidylinositol anchor [7]. The similar effects of brinzolamide and dorzolamide on CBF suggest that the CAII is involved in the regulation of pH_i_ and CBF in c-hNECS, as reported in the bicarbonate transport metabolon [7,8,12].

##### Effects of NBC Inhibitors (S0859 and DIDS) on CBF and pH_i_ in c-hNECs

The present study suggests that a high pH_i_ appears to cause the right shift of Equation (1) in c-hNECs. A high pH_i_ may be produced by the HCO_3_^−^ entry via NBC in c-hNECs. To inhibit NBC, S0859 (30 μM, a selective inhibitor of NBC) was used. The addition of S0859 decreased the CBF ratio from 1.00 to 0.79 (*n* = 9, at 5 min after the addition). Removing S0859 recovered the CBF ratio to 0.99 (at 1 min after the removal) (Figure 8A). Experiments were also carried out using DIDS (200 µM, an inhibitor of NBC and AE) (Figure 8B). The addition of DIDS, similarly to S0859, decreased the CBF ratio from 0.99 to 0.78 (*n* = 5, at 5 min after the addition). However, the removal of DIDS did not recover the CBF ratio, and the CBF ratio was 0.71 at 5 min after the removal. The pH_i_ was measured. S0859 decreased the pH_i_ in c-hNECs, and the pH_i_ values were 7.54 before and 7.45 (*n* = 10, *p* < 0.01) at 5 min after the addition of S0859. The removal of S0859 immediately increased the pH_i_ and then decreased gradually. The pH_i_ was 7.60 at 3 min after the removal (Figure 8C). Thus, the inhibition of NBC decreases pH_i_ and CBF in c-hNECs.

The effects of brinzolamide and S0859 on CBF were examined in c-hNECs. In the presence of brinzolamide, S0859 did not decrease the CBF ratio (Figure 9A), suggesting that the interactions of CAII with NBC may function in the bicarbonate transport in c-hNECs [7,8,12]. In c-hNECs treated with S0859 for 1 h, the application of Zero-CO_2_ increased the CBF ratio transiently (Figure 9B). The CBF ratio was 1.00 (*n* = 4) before and 1.19 at 3 min after applying the Zero-CO_2_. This suggests that application of the Zero-CO_2_ increased pH_i_ in S0859-treated c-hNECs. Inhibition of HCO_3_^−^ entry by S0859 appears to increase pH_i_ in c-hNECS, which shifts Equation (1) to the left upon applying the Zero-CO_2_. These results suggest that the interactions of CAII and CAIV with NBC accelerate the rate of HCO_3_^−^ influx in c-hNECs. A high rate of HCO_3_^−^ entry through NBC causes an increase in the pH_i_ to an extremely high level in c-hNECs. Previous studies have demonstrated that CAIV plays a key role in maximizing the rate of HCO_3_^−^ transport via NBC in kidney proximal tubules and retinal epithelia [7,8,9,10,12].

#### 2.3.2. Ciliated hBECs

To clarify the role of CAIV in the HCO_3_^−^ transport, similar experiments were carried out using c-hBECs expressing no CAIV. Ciliated hBECs were differentiated from NHBE cells by the ALI culture [13]. We measured CBF and pH_i_ upon applying Zero-CO_2_ and the NH_4_^+^ pulse in c-hBECs.

##### Effects of Zero-CO_2_ and the NH_4_^+^ Pulse on CBF and pH_i_ in c-hBECs

Application of Zero-CO_2_ transiently increased CBF and CBD in c-hBECs. The CBF and CBD ratios at 2 min after the application were 1.10 (*n* = 6) and 1.18 (*n* = 5), respectively (Figure 10A). Changes in the pH_i_ were also measured. The pH_i_ was 7.10 (*n* = 10) in c-hBECs perfused with the control solution. The application of Zero-CO_2_ transiently increased pH_i_ (Figure 10B). Changes in the CBF ratio and pH_i_ during the NH_4_^+^ pulse were measured. Application of the NH_4_^+^ pulse increased and plateaued the CBF ratio within 2 min (Figure 10C). Application of the NH_4_^+^ pulse increased and plateaued the pH_i_ (Figure 10D). The pH_i_ before the NH_4_^+^ pulse was 6.96 (*n* = 8). The NH_4_^+^ pulse immediately increased the pH_i_, and the pH_i_ at 5 min from the start of the NH_4_^+^ pulse was 7.26 (*n* = 10). The pH_i_ of c-hBECs was lower than that of c-hNECs. The application of Zero-CO_2_ transiently increased the CBF ratio and pH_i_, and the application of the NH_4_^+^ pulse increased and plateaued the CBF ratio and pH_i_ in c-hBECs (Figure 10).

##### Effects of Brinzolamide and S0859 on CBF and pH_i_ in c-hBECs

Brinzolamide (0.1 µM) gradually decreased the CBF ratio in c-hBECs, suggesting that the left shift of Equation (1) occurred via CAII. The CO_2_ may be synthesized via CAII from H^+^ produced by the cellular metabolism and other systems during the control perfusion. Application of the NH_4_^+^ pulse increased and plateaued the CBF ratio. Cessation of the NH_4_^+^ pulse immediately decreased the CBF ratio (Figure 11A). Changes in pH_i_ were measured. The addition of brinzolamide slightly decreased pH_i_ (not significant). Application of the NH_4_^+^ pulse sustained pH_i_ (Figure 11B). The pH_i_ values before and 5 min after the brinzolamide addition were 7.43 and 7.38 (*n* = 5), respectively, and the pH_i_ at 5 min from the start of the NH_4_^+^ pulse was 7.77. The addition of S0859 did not change the CBF ratio, but the removal of S0859 gradually increased the CBF ratio (Figure 11C). NBC appears to function in c-hBECs. Changes in pH_i_ were measured. S0859 did not change the pH_i_, but the removal of S0859 increased the pH_i_ slightly (Figure 11D). These suggest that pH_i_ is maintained by interactions of CAII and NBC in c-hBECs, and the increases in CBF and pH_i_ after removing S0859 indicate that HCO_3_^−^ enters cells through NBC during control perfusion. The effects of S0859 on CBF and pH_i_ indicate that the rate of HCO_3_^−^ influx is much lower in c-hBECs than in c-hNECs.

## 3. Discussion

The present study demonstrated that the pH_i_ of c-hNECs is extremely high (7.66), and the high pH_i_ is generated by a high rate of HCO_3_^−^ influx in c-hNECs. The [HCO_3_^−^]_i_ is calculated to be 41.4 mM from pH_i_ (7.66) and pCO_2_ (5% CO_2_, 38 mmHg) by the Henderson–Hasselbalch equation in c-hNECs. This study also demonstrated that the pH_i_ of c-hBECs is low (7.10), and the low pH_i_ is generated by a low rate of HCO_3_^−^ influx. The [HCO_3_^−^]_i_ is calculated to be 11.4 mM from the pH_i_ (7.1) and pCO_2_ (38 mmHg) in c-hBECs. The [HCO_3_^−^]_i_ of c-hNECs is approximately four times higher than that of c-hBECs.

The present study revealed that the application of Zero-CO_2_ decreases pH_i_ mediated via decreases in [HCO_3_^−^]_i_ due to no HCO_3_^−^ entry. An extremely high pH_i_ (a low [H^+^]_i_) and a low [HCO_3_^−^]_i_ appear to induce the right shift (H^+^ production) or no shift (no elimination of H^+^) of Equation (1) even in Zero-CO_2_, in which a small amount of H^+^ is supplied from the cellular metabolism. A decrease in pH_i_ caused CBF and CBD to decrease in c-hNECs [1,2,4]. However, in the c-hBECs, the pH_i_ and [HCO_3_^−^]_i_ were low because of a low rate of HCO_3_^−^ influx. The switch to the Zero-CO_2_ from the control solution immediately removes CO_2_ from the extracellular space to decrease CO_2_ concentration ([CO_2_]_i_), keeping a low pH_i_ in c-hBECs. The low pH_i_ and lowered [CO_2_]_i_ shift Equation (1) to the left to increase pH_i_ in c-hBECs. The pH_i_ increase enhances CBF and CBD in c-hBECs [1,2,4]. Thus, a low pH_i_ shifts Equation (1) to the left upon applying the Zero-CO_2_ in c-hBECs.

The application of Zero-CO_2_ appears to induce a large decrease in [HCO_3_^−^]_i_ in c-hNECs. The effects of the decrease in [HCO_3_^−^]_i_ on Equation (1) may be much larger than those of the decrease in [CO_2_]_i_ in c-hNECs upon applying Zero-CO_2_. In c-hBECs, however, the application of Zero-CO_2_ decreases [CO_2_]_i_ to an extremely low level and may induce little decrease in [HCO_3_^−^]_i_, because of a low HCO_3_^−^ influx rate. The effects of the decrease in [CO_2_]_i_ on Equation (1) may be much larger than those of the [HCO_3_^−^]_i_ decrease in c-hBECs, to induce the left shift upon applying Zero-CO_2_.

The high rate of HCO_3_^−^ transport into cells is maintained in c-hNECs, leading to an extremely high pH_i_. RT-PCR analysis revealed that five NBC subtypes (NBC1, NBCe1, NBCe2, NBCn1 and NDCBE) and AE (SLC4A2 (AE2)) are expressed in both c-hNECs and c-hBECs. The present study demonstrated that NBC blockers (S0859 and DIDS) decrease CBF in c-hNECs, but they do not change CBF in c-hBECs. Thus, the activity of NBC is high in c-hNECs but not in c-hBECs. This indicates that the mechanism stimulating NBC activity exists in c-hNECs.

The contribution of AEs to the HCO_3_^−^ entry appears to be small, because there was no difference between CBFs decreased by S0859 and those decreased by DIDS in c-hNECs. In c-hNECs, the pH_i_ is high, except in some experimental conditions, such as the long-time exposure to Zero-CO_2_ and the removal of the NH_4_^+^ pulse. Under these experimental conditions, the Na^+^/H^+^ exchange (NHE) may extrude H^+^ from c-hNECs. However, the NHE is unlikely to increase the pH_i_ to an extremely high level in c-hNECs during the ALI culture, because it has been shown to be inactive at pH_i_ levels higher than 7.4 [18].

The present study demonstrated that CAIV is expressed in c-hNECs but not in c-hBECs. The expression of CAIV has already been shown in human nasal epithelia [7]. A previous study demonstrated that the physical and functional interactions between CAIV and NBC maximize transmembrane HCO_3_^−^ transport in HEK239 cells transfected with NBC1b and CAIV [8], renal proximal tubules [9] and a retinal photoreceptor, which has been detected in an autosomal dominant form of retinitis pigmentosa (the R14W mutation of CAIV) [10]. These findings suggest that CAIV interacts with NBC to increase the activity of the HCO_3_^−^ transporter in the c-hNECs. Ciliated hBECs express HCO_3_^−^ transporters but no CAIV. A previous report showed that the expression of CAIV was low in the trachea [19]. The NBC blocker study showed that the activity of NBC is low in c-hBECs, as described above. Moreover, c-hBECs kept a low pH_i_. These indicate that no expression of CAIV causes a low activity of NBC in c-hBECs. These results indicate that CAIV increases the NBC activity to maximize the rate of HCO_3_^−^ transport into cells in c-hNECs.

CAII has been shown to interact with NBC1 [12,20]. CAII and CAIV have similar structures, and the acid motif in the NBC1 C-terminal region interacts with the basic N-terminal region of CAII [12,20]. The HCO_3_^−^s are produced by CAIV in the apical surface, entering the cell via the NBCs, and the HCO_3_^−^ entered is converted to CO_2_ by CAII just below the apical membrane. The coupling of CAIV-NBC-CAII appears to potentiate transmembrane HCO_3_^−^ influxes in c-hNECs [8,12]. In this study, brinzolamide (CAII inhibitor) and dorzolamide (CAII and CAIV inhibitor) showed similar decreases in CBF and pH_i_. These results suggest that the interactions of CAIV and CAII with NBC may potentiate the influx of HCO_3_^−^ in c-hNECs. The CAIV, NBC and CAII have been shown to compose the bicarbonate transport metabolon in renal proximal tubules [7,8,9,12]. The c-hNECs may also express the bicarbonate transport metabolon consisting of CAIV-NBC-CAII, which maximizes the rate of HCO_3_^−^ transport from the apical surface into the cell. However, we do not confirm that the metabolon consisted of CAIV-NBC-CAII in c-hNECs. Further experiments are needed.

A membrane-bound CA, CAIX, is expressed in c-hNECs and c-hBECs. The CAIX mRNA expression has been shown in the apical surface of nasal mucosa [6]. CAIX has been shown to interact with AE2 in HEK293 cells, co-expressing the parietal cell AE2 and CAIX, and increases the activity of AE2 transport by 28% [21]. However, the interactions between CAIX and NBC remain uncertain. The present study revealed that CAIX is also expressed in c-hBECs, the HCO_3_^−^ entry is not enhanced in c-hBECs and the HCO_3_^−^ entry via AE appears to be small in c-hNECs. Based on these observations, the CAIX is unlikely to enhance NBC in c-hNECs.

The present study does not provide the localization of NBC isoforms in the apical membrane of c-hNECs. However, it has been demonstrated that NBC1 and CAIV have a physical and functional relationship in HEK293 cells transfected with NBC1 and CAIV [7,8,12]. Liu et al. also showed that NBC functionally exists in apical membranes of mice bronchioles [17]. The present study suggests that NBC1 exists in the apical membrane of c-hNECs to form the bicarbonate transport metabolon.

The application of Zero-CO_2_ induced various responses in CBF and pH_i_, a decrease (Figure 5) or no change (Figure 6A), although it never induced large increases as shown in c-hBECs (Figure 10) or bronchial ciliated cells [1]. The responses of c-hNECs were affected by cellular conditions. In this study, c-hNECs with a permeable support filter were kept in the control solution without any aeration at room temperature until the measurement of CBF or pH_i_. This condition may decrease the HCO_3_^−^ entry and may induce a small decrease in the pH_i_ of c-hNECs, which decreases CBF.

The HCO_3_^−^ entry appears to be affected by temperature-keeping cells and time-keeping cells until the start of the experiments, depending on the conditions keeping the cells until the experiments, such as time and temperature. After keeping c-hNECs at 4 °C for more than 3 h, the pH_i_ decreased by approximately 0.1–0.15, and the application of Zero-CO_2_ induced no changes or small increases in CBF.

In this study, we used two nasal tissues (uncinate process and nasal polyp tissues resected from 16 patients who required surgery for CS). We measured CBF in c-hNECs differentiated from the two nasal tissues. The CBFs of c-hNECs obtained from the two nasal tissues were similar. Moreover, c-hNECs obtained from the two nasal tissues expressed the CAIV mRNA. Kim et al. demonstrated that the expression levels of eleven CA isozymes were decreased by 80–40% in the nasal polyp tissue [6]. However, they examined mRNA expression using whole samples, and expression of CA isozymes in nasal polyp tissue was weak in the epithelial layer but weaker or absent in the submucosal glands and vascular endothelial cells. We used c-hNECs cultured by ALI, which contain no submucosal gland and no vascular endothelial cells. Based on these observations, CAII and CAIV, at least, express and function in c-hNECs obtained from nasal polyp samples, although the expression level may be lower than in normal nasal epithelia.

We used c-hBECs obtained by the ALI culture from NHBEs as a model of human tracheal epithelia. The NHBEs were bought from Lonza (Lot No. 20TL119094). The c-hBECs used appear to be a good model of tracheal ciliated epithelial cells.

Ciliated-hNECs were cultured in the ALI with 5% CO_2_ for more than 4 weeks. The culture condition with 5% CO_2_ is different from the asymmetrical gas condition of c-hNECs and c-hBECs in vivo; the apical surface is exposed to the air (0.04% CO_2_) periodically, and the basolateral membranes are exposed to interstitial fluid saturated with 5% CO_2_ (Figure 12). The ALI culture with 5% CO_2_ enhances the HCO_3_^−^ transport into c-hNECs. This unphysiological gas condition appears to increase pH_i_ to an extremely high level by maximizing the HCO_3_^−^ transport via the interactions of CAIV, NBC and CAII (Figure 12A). In c-hBECs expressing no CAIV, CO_2_ is converted to H^+^ and HCO_3_^−^ by CAII. The H^+^ produced stays in the c-hBECs to decrease pH_i_, while HCO_3_^−^ is secreted to the lumen via CFTR and AE [17]. Thus, the ALI culture condition may enhance HCO_3_^−^ entry, leading to a high pH_i_ in c-hNECs, but it may also enhance the conversion of CO_2_ to H^+^, leading to a low pH_i_ in c-hBECs.

In conclusion, we found novel interactions with CAIV, NBC and CAII, which regulate pH_i_ in c-hNECs. CAIV, NBC and CAII may consist of a bicarbonate transport metabolon in c-hNECs. In the physiological condition, CO_2_ diffuses to the apical surface from the interstitial space according to the CO_2_ gradient between the interstitial fluid (5%) and the nasal cavity (0.04–1%). The CO_2_ leaked is converted to H^+^ and HCO_3_^−^ by CAIV around the cilia. The HCO_3_^−^ enters cells via NBC, and the H^+^ stays in the nasal surface mucous layer to keep a low pH. The low pH of the nasal mucous layer is essential for the protection from inhaled bacteria [22,23] (Figure 12B). HCO_3_^−^ entered via NBC immediately coverts to CO_2_ by CAII. The removal of HCO_3_^−^ by CAII enables the transportation of HCO_3_^−^ continuously into the cell by keeping the driving force for HCO_3_^−^ entry through NBC. Although we do not know the exact pCO_2_ and [HCO_3_^−^]_i_ of c-hNECs, the HCO_3_^−^ transport metabolon appears to be essential for maintaining the pH_i_ and the ciliary beating of c-hNECs at adequate levels in the nasal cavity with low CO_2_ concentrations (0.04–1%). The novel mechanism accelerating bicarbonate transport via the HCO_3_^−^ transport metabolon (CAIV-NBC-CAII) appears to play essential roles in protecting nasal mucosa from inhaled small particles, such as bacteria, virus and chemicals, and maintaining healthy nasal mucosa. Further studies are required to understand this novel mechanism in nasal epithelia in vivo.

## 4. Materials and Methods

### 4.1. Ethical Approval

This study has been approved by the ethical committees of the Kyoto Prefectural University of Medicine (RBMR-C-1249-7) and Ritsumeikan University (BKC-HM-2020-090). All experiments were performed according to the ethical principles for medical research outlined in the Declaration of Helsinki (1964) and its subsequent revisions (https://www.wma.net/, accessed on 1 April 2020). Informed consents were obtained from all patients before operation. Human nasal tissue samples (nasal polyp or uncinate process) were resected from patients who required surgery for chronic sinusitis (16 patients). Samples were immediately cooled and stored in the cooled control solution (4 °C) until cell isolation [2].

### 4.2. Solution and Chemicals

The control solution contained (in mM) NaCl 121, KCl 4.5, NaHCO_3_ 25, MgCl_2_ 1, CaCl_2_ 1.5, NaHEPES 5, HHEPES 5 and glucose 5. Its pH was adjusted to 7.4 by HCl (1 M), and the solution was aerated with 95% O_2_ and 5% CO_2_. The CO_2_/HCO_3_^−^-free control solution was prepared by replacing NaHCO_3_ in the control solution with NaCl and was aerated with 100% O_2_. To apply the NH_4_^+^ pulse, the NaCl (25 mM) of the solutions was replaced with NH_4_Cl (25 mM). DNase I, amphotericin B, DIDS (4,4-diisothiocyanatostilbene-2,2-disulfonic acid disodium salt hydrate) and S0859 (a selective NBC inhibitor, 2-chloro-N-((2′-(N-cyanosulfamoyl)-[1,1′-biphenyl]-4-yl)methyl)-N-(4-methylbenzyl) benzamide) were purchased from Sigma-Aldrich (St Louis, MO, USA). Dorzolamide and brinzolamide were purchased from Tokyo Chemical Industry Co., Ltd. (Tokyo, Japan). The Can Get Signal^®^ Immunoreaction Enhancer Solution was purchased from TOYOBO (Osaka, Japan).

### 4.3. Cell Culture Media

The complete PneumaCultTM-Ex Plus medium contained PneumaCultTM-Ex Plus basal medium supplemented with PneumaCultTM-Ex Plus supplement (50×, 20 µL/mL), hydrocortisone stock solution (1 µL/mL) and penicillin and streptomycin solution (10 µL/mL). The complete PneumaCultTM-ALI medium contained PneumaCultTM-ALI basal medium supplemented with PneumaCultTM-ALI supplement (10×, 100 µL/mL), PneumaCultTM-ALI maintenance supplement (10 µL/mL), heparin solution (2 µL/mL), hydrocortisone stock solution (2.5 µL/mL) and penicillin/streptomycin solution (10 µL/mL). Solutions and supplements were purchased from STEMCELL Technologies, Inc. (Vancouver, BC, Canada). Elastase, bovine serum albumin (BSA) and dimethyl sulfoxide (DMSO) were purchased from FUJIFILM Wako Pure Chemical Corporation (Osaka, Japan). Penicillin/streptomycin mixed solution (penicillin 10,000 units/mL and streptomycin 10,000 µg/mL in 0.85% NaCl), trypsin, and the trypsin inhibitor were purchased from Nacalai Tesque, Inc. (Kyoto, Japan).

### 4.4. Antibodies

The anti-CAIV antibody (AF2186, polyclonal goat antibody) was purchased from R&D Systems (Minneapolis, MN, USA). The concentration of AF2186 used was 1 µg/mL. The antigen peptide (2186-CA, recombinant human CAIV) was also purchased from R&D systems. The anti-alpha-tubulin (acetyl K40) (AC-tubulin) antibody (ab179484) was purchased from Abcam plc (Cambridge, UK) and used at a 100-fold dilution. Alexa Fluor 488 goat anti-mouse IgG (H+L) secondary antibodies (A-11001) and Alexa Fluor 594 donkey anti-rabbit IgG (H+L) secondary antibodies (A-21207) were purchased from Thermo Fischer Scientific (Waltham, MA, USA).

### 4.5. Cell Preparation

We isolated c-hNECs from nasal operation samples as described previously [2]. Briefly, resected samples were cut into small pieces and incubated for 40 min at 37 °C in a control solution containing elastase (0.02 mg/mL), DNase I (0.02 mg/mL) and BSA (3%). Then, the samples were minced in a control solution containing DNase I (0.02 mg/mL) and BSA (3%) using fine forceps. Isolated nasal cells were washed with a control solution containing BSA (3%) three times with centrifugation at 160× *g* for 5 min and then sterilized for 15 min using amphotericin B (0.25 μg/mL) in Ham’s F-12 with L-glutamine. Isolated nasal epithelial cells were cultured in complete PneumaCult-Ex Plus medium in a collagen-coated flask (Corning, 25 cm^2^, New York, NY 14831 USA) at 37 °C in a humidified 5% CO_2_ atmosphere. The medium was changed every second day. Once the cells reached confluency, they were washed with PBS (5 mL) and harvested in Hank’s balanced salt solution (HBSS, 2 mL) containing 0.1 mM EGTA and 0.025% trypsin to remove cells from the flask. Then, a trypsin inhibitor was added into the cell suspension to stop further digestion. After washing with centrifugation, cells were resuspended in complete PneumaCultTM-Ex Plus medium (1–2 × 10^5^ cells, 3 mL) and seeded on a filter of Transwell permeable supports insert (Coster 3470, 6.5 mm Transwell with 0.4 μm Pore Polyester Membrane Inserts, Corning) (3.0 ×10^4^ cells/insert, 400 μL). The complete PneumaCultTM-Ex Plus medium was added into the upper and bottom chambers, and the cells were cultured until confluent. Then, the medium in the bottom chamber was replaced with the complete PneumaCultTM-ALI medium (500 μL), and the medium in the upper chamber was removed to expose cells to the air (ALI culture). The medium in the bottom chamber was changed thrice per week. Cells were cultured for 4 weeks under the ALI condition to allow differentiation into ciliated cells [5].

NHBE cells were purchased from Lonza (LOT No. 20TL119094, Basel, Switzerland) and cultured in the flask, in which complete PneumaCultTM-Ex Plus medium was added at 37 °C in a humidified 5% CO_2_ atmosphere. Once the cells had reached confluency, they were washed with PBS (5 mL) and harvested with HBSS (2 mL) containing 0.1 mM EGTA and 0.025% trypsin. Then, a trypsin inhibitor was added. After washing the cells with centrifugation, the cells were resuspended in complete PneumaCultTM-Ex Plus medium (3 mL). The cells were seeded onto the filter of the Transwell permeable supports inserts (3.0 × 10^4^ cells/insert, 400 μL) and cultured into the complete PneumaCultTM-Ex Plus medium, which was also added to the upper and bottom chambers. Once the cells reached confluency, the medium in the bottom chamber was replaced with PneumaCultTM-ALI medium (500 μL), and the medium in the upper chamber was removed (ALI culture). The medium in the bottom chamber was changed thrice per week. Cells were cultured for 3 weeks under the ALI condition [18]. There were no differences in the development of cilia between nasal epithelial and NHBE cells.

### 4.6. Measurements of CBF and CBD

The insert membrane filter, on which cells had grown, was cut into 4–6 pieces. A piece of membrane with cells was placed on a coverslip precoated with neutralized Cell-Tak (Becton Dickinson Labware, Bedford, MA, USA). The coverslip with cells was then set in a perfusion chamber (20 µL), which was mounted on an inverted microscope (T-2000, NIKON, Tokyo, Japan) connected to a high-speed camera (IDP-Express R2000, Photron Ltd., Tokyo, Japan) (high-speed video microscope) [2,24]. The cells were perfused at a constant rate (200 µL/min). Since CBF is sensitive to temperature, the experiments were carried out at 37 °C [2,4,5,24]. Video images were recorded for 2 s at 500 fps using a high-speed video microscope. Video images of c-NECs before and 5 min after applying the NH_4_^+^ pulse are shown in Appendix A, respectively. The methods to measure CBF and CBD (ciliary bend distance, an index of ciliary beating amplitude (CBA)) have been described in detail [5,6,8,9]. The ratios of CBF (CBF_t_/CBF_0_) and CBD (CBD_t_/CBD_0_) were calculated to make comparisons across the experiments. The subscripts ‘0’ and ‘t’ indicate the time from the start of the experiments. Cells with the cut filter were kept in the control solution at room temperature (2–3 h) until the start of the CBF and CBD measurements. The storage conditions, such as temperature and time, affected CBF responses upon Zero-CO_2_ application with a decrease, no change or a small increase, as shown in Figure 5 and Figure 6.

### 4.7. Measurement of pH_i_

The insert membrane filter with cells was incubated with a Ca^2+^-free control solution containing 1 mM EGTA (pH 7.2) for 10 min at room temperature, and then the cell sheet was removed from the membrane filter using a fine forceps. Then, the cell sheet was incubated with 2 µM BCECF-AM (Dojindo Laboratories, Kumamoto, Japan) for 30 min at 37 °C. After BCECF loading, the cell sheet was cut into small pieces (4–6 pieces) and kept in the control solution at room temperature until pH_i_ measurements. A piece of cell sheet was set in a perfusion chamber, and the fluorescence of BCECF was measured using an image analysis system (MetaFluor, Molecular Device, CA, USA). BCECF was excited at 440 nm and 490 nm, and the emission was recorded at 530 nm. The fluorescence ratio (F490/F440) was calculated and recorded by the image analysis system. The calibration curve for pH_i_ was obtained using BCECF-loaded cells perfused with a calibration solution containing nigericin (15 µM, Sigma-Aldrich, St Louis, MO, USA). The pHs of the calibration solution were 6.5, 7.0, 7.5 and 8.0. The calibration solution contained (in mM) KCl 150.5, MgCl_2_ 2, CaCl_2_ 1, HEPES 10 and glucose 5.

### 4.8. RT-PCR

Total RNA samples from c-hNECs and c-hBECs were prepared using an RNeasy Minikit (QIAGEN, Tokyo, Japan). Total RNA was reverse transcribed to cDNA using an oligo d(T)6 primer and an Omniscript RT kit (QIAGEN). Then, cDNA samples were subjected to reverse transcription-polymerase chain reaction (RT-PCR) using KOD FX (TOYOBO). The gene-specific primers for human CA are listed in Table 1, and those for human NBC and anion exchangers (AE) are in Table 2. The amplified PCR products were confirmed using agarose gel.

Real-time PCR was performed in c-hNECs and c-hBECs using the cDNA and CAIV primers confirmed by RT-PCR, and the expression levels of CAIV mRNA were quantitatively evaluated. Quantitative analyses for CAIV mRNA expression and GAPDH mRNA expression were performed using the PowerUp SYBR Green Master Mix (Applied Biosystems, Waltham, MA, USA). The expression level of CAIV mRNA was normalized to that of GAPDH.

### 4.9. Western Blotting

Cells on the insert membrane filter were washed with PBS and removed from the filter. Then, cells were homogenized in a radioimmunoprecipitation assay buffer (50 mM Tris-HCl, 150 mM NaCl, 1% Nonidet-P40, 0.5% sodium deoxycholate and 0.1% SDS, pH 7.6) containing a protease inhibitor cocktail and incubated at 4 °C for 20 min. Cells were then centrifuged at 16,000× *g* for 20 min at 4 °C. The supernatant was used as a cell lysate. The lysate was incubated with PNGase F (a recombinant glycosidase, Roche, Basel, Switzerland) in PBS containing 15 mM EDTA, 1% Nonidet P-40, 0.2% SDS and 1% 2-mercaptoethanol at 37℃ overnight. Proteins were separated using Laemmli’s SDS-polyacrylamide gel electrophoresis (8–12.5%) and then transferred onto a polyvinylidene difluoride membrane. The membrane was blocked with milk (2.5%) in Tris-buffered saline (10 mM Tris-HCl and 150 mM NaCl, pH 8.5) containing 0.1% Tween 20 (TBST) for 1 h and then incubated with a primary antibody (MAB2186, R&D System) diluted in solution 1 (Can Get Signal Immunoreaction Enhancer Solution, TOYOBO) overnight at 4 °C. After washing with TBST, the membrane was incubated with a secondary antibody (AP124P, anti-mouse IgG) diluted in solution 2 (Can Get Signal Immunoreaction Enhancer Solution, TOYOBO) for 1 h at room temperature. After washing, antigen–antibody complexes on the membrane were visualized using a chemiluminescence system (ECL plus; GE Healthcare, Waukesha, WI, USA).

### 4.10. Immunofluorescence Examination

Immunofluorescence examinations were performed in c-hNECs and c-hBECs [19]. The cells on the Transwell insert membrane filter were removed using a cell scraper and suspended in PBS (2 mL). The cell suspension (0.5 mL) was dropped and dried on the cover slip, to which the cells attached. Then, the cells were fixed in 4% paraformaldehyde for 30 min and washed three times with PBS containing 10 mM glycine. The cells were permeabilized with 0.1% Triton X-100 for 15 min at room temperature. After 60 min pre-incubation with PBS containing 3% BSA at room temperature, the cells were incubated with the anti-CAIV (AF2186) and anti-AC-tubulin (ab179484, Abcam) antibodies overnight at 4 °C. Then, the cells were washed with PBS containing 0.1% BSA to remove unbound antibodies. Afterwards, the cells were stained with Alexa Fluor 488 goat anti-mouse IgG (H+L) (A-11001, 1:100 dilution) and Alexa Fluor 594 donkey anti-rabbit IgG (H+L) (A-21207, 1:100 dilution) secondary antibodies for 60 min at room temperature. The samples on the coverslip were enclosed with a mounting medium with DAPI (Vector, Burlingame, CA, USA). The cells were observed using a confocal microscope (LSM900, ZEISS, Oberkochen, Germany) [4,5].

### 4.11. Statistical Analysis

Statistical significance was assessed using one-way analysis of variance or Student’s *t*-test (paired or unpaired), as appropriate. Differences were considered significant for *p*-values < 0.05. The results are expressed as the means ± SD.

## 5. Conclusions

The interactions of CAIV, NBC and CAII, which appear to consist of the bicarbonate transport metabolon, maximized the rate of transmembrane HCO_3_^−^ transport into the cell in c-hNECs. A large amount of the HCO_3_^−^ entered traps H^+^ to increase pH_i_ to an extremely high level. The high level of pH_i_ prevents the CBF increase upon applying Zero-CO_2_. In the physiological condition, the apical surface is periodically exposed to the air (0.04% CO_2_) with the respiration, while the basolateral membrane was faced to the interstitial fluid saturated with 5% CO_2_. In c-hNECs exposed to the asymmetrical CO_2_ condition, the bicarbonate transport metabolon appears to keep an adequate pH_i_ and ciliary beating. The active beating cilia are essential for the active mucociliary clearance removing inhaled pathogens [25,26]. This mechanism maintains a healthy nasal cavity.

## Figures and Tables

**Figure 1 ijms-25-09069-f001:**
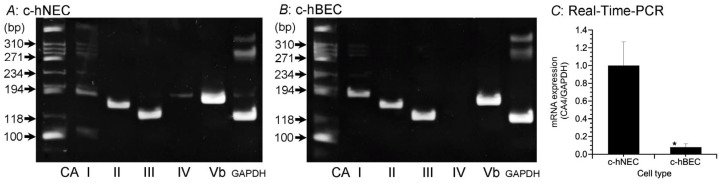
Expression of CA subtypes’ mRNAs. The expressions of CAI, CAII, CAIII, CAIV and CAVb were examined using RT-PCR. (**A**) c-hNECs. The mRNAs of five CA subtypes were expressed. (**B**) c-hBECs. The mRNAs of four CA subtypes with the exception of CAIV were expressed. CAIV was expressed in c-hNECs, but not in c-hBECs. (**C**) The real-time PCR examination of CAIV mRNA in c-hNECs and c-hBECs. The expression level of CAIV mRNA was significantly higher in c-hNEC than in c-hBECs. * significantly different (*p* < 0.01).

**Figure 2 ijms-25-09069-f002:**
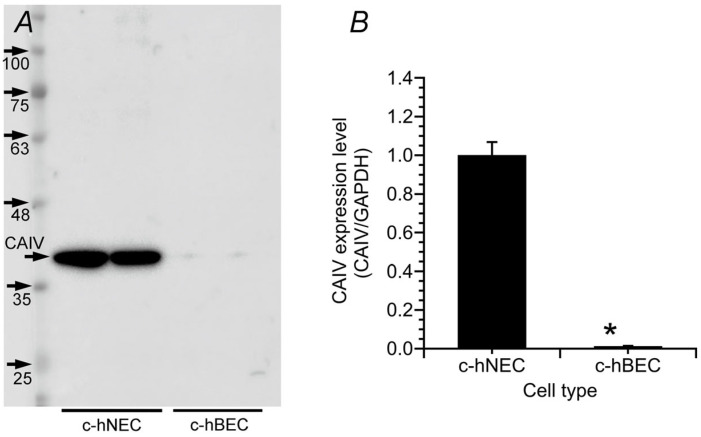
Western blotting for CAIV in c-hNECs and c-hBECs. (**A**) The single band of CAIV (40 kDa) was detected in c-hNECs but faint in c-hBECs. (**B**) Densitometric analysis. The expression of CAIV protein was higher in c-hNECS (*n* = 4) than in c-hBECs (*n* = 4). * significantly different (*p* < 0.01).

**Figure 3 ijms-25-09069-f003:**
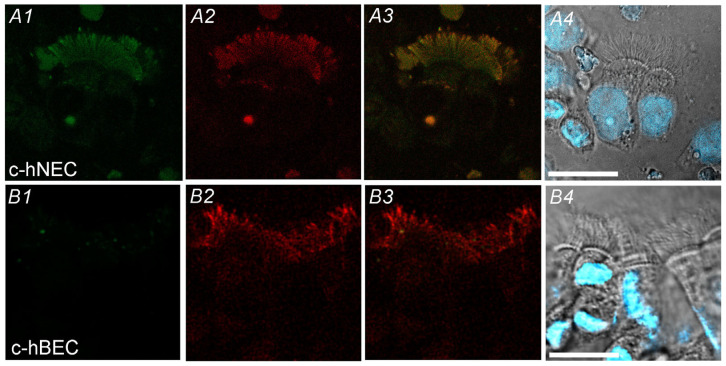
Immunofluorescence examination for CAIV. (**A1**–**A4**) c-hNEC. (**A1**): CAIV (green), (**A2**): AC-tubulin (a cilia marker, red), (**A3**): merged image, (**A4**): phase contrast image and nuclei were stained by DAPI (blue). The cilia of c-hNEC were immunopositively stained for CAIV. (**B1**–**B4**) c-hBEC. (**B1**): CAIV, (**B2**): AC-tubulin, (**B3**): merged image, (**B4**): phase contrast image with DAPI staining. No cilia of c-hBECs were stained for CAIV. Scale bar 20 µm.

**Figure 4 ijms-25-09069-f004:**
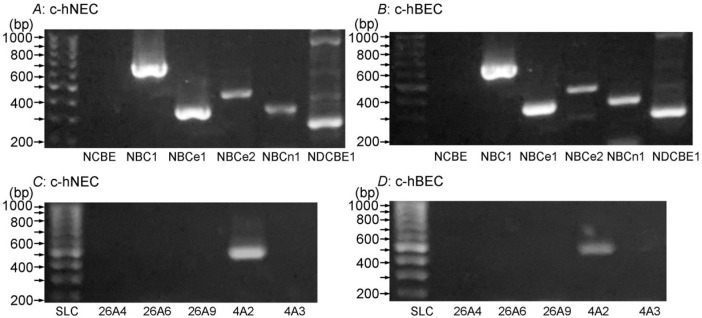
NBC expression (**A**,**B**) and AE expression (**C**,**D**) examined by RT-PCR in c-hNECs and c-hBECs. (**A**,**C**) c-hNECS. (**B**,**D**) c-hBECs. There is no difference in the mRNA expression of NBCs or AEs between c-hNECs and c-hBECs.

**Figure 5 ijms-25-09069-f005:**
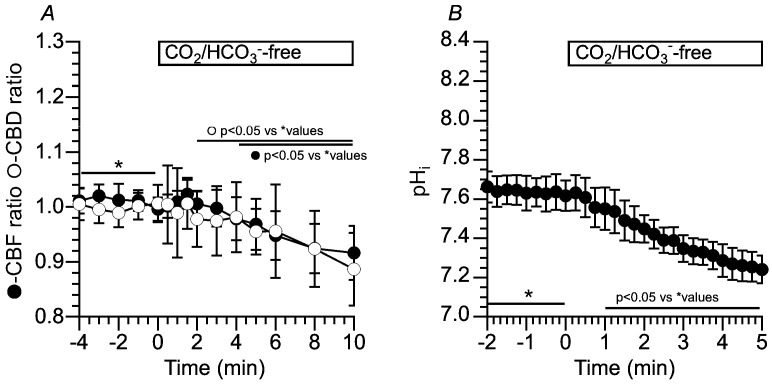
Changes in CBF, CBD and pH_i_ in c-hNECs upon applying Zero-CO_2_. (**A**) Changes in CBF and CBD ratios. Application of Zero-CO_2_ induced a small transient increase followed by a gradual decrease in CBF or CBD. (**B**) Changes in pH_i_. Application of Zero-CO_2_ gradually decreased pH_i_ from 7.65 to 7.30. * shows control values.

**Figure 6 ijms-25-09069-f006:**
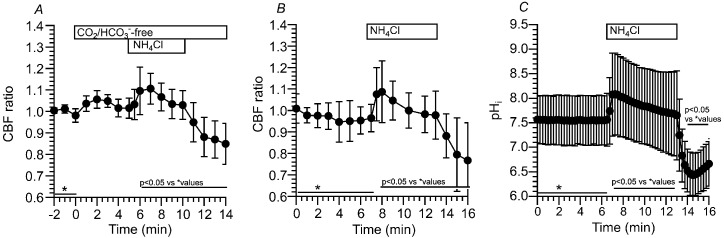
Changes in CBF and pH_i_ induced by the NH_4_^+^ pulse in c-hNECs. (**A**) Changes in CBF upon applying the NH_4_^+^ pulse in the Zero-CO_2_. The application of Zero-CO_2_ did not increase CBF. The NH_4_^+^ pulse induced an immediate increase followed by a gradual decrease in CBF. The cessation of the NH_4_^+^ pulse immediately decreased CBF. (**B**) Changes in CBF induced by the NH_4_^+^ pulse in the control solution. The application of NH_4_^+^ pulse induced an immediate increase followed by a gradual decrease in CBF. Cessation of the NH_4_^+^ pulse immediately decreased CBF. (**C**) Changes in pH_i_ upon applying the NH_4_^+^ pulse. The NH_4_^+^ pulse induced an immediate increase followed by a gradual decrease in pH_i_. Cessation of the NH_4_^+^ pulse immediately decreased pH_i_, and then the pH_i_ gradually increased to the control level. * shows control values.

**Figure 7 ijms-25-09069-f007:**
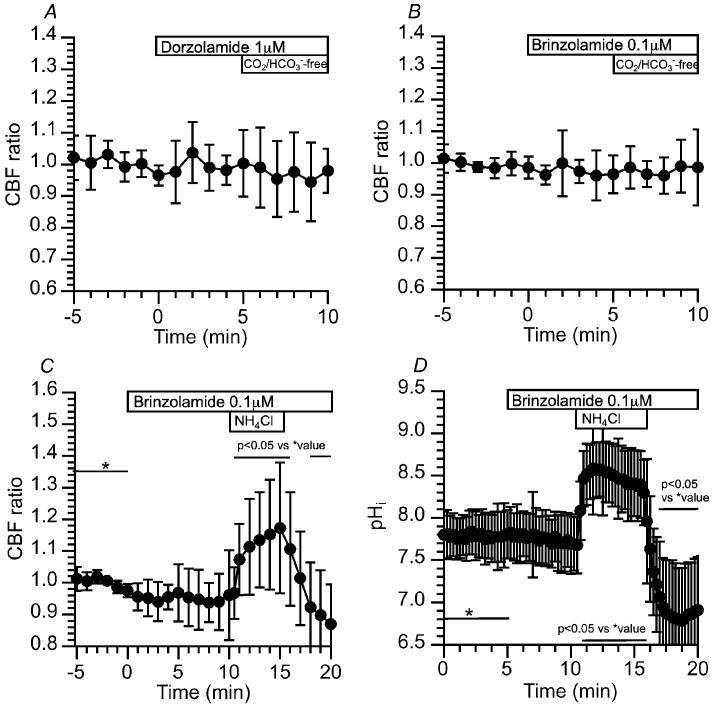
Effects of dorzolamide (an inhibitor of CAII and CAIV) and brinzolamide (a selective inhibitor of CAII) on CBF and pH_i_ in c-hNECs. (**A**) Effects of dorzolamide (1 µM) on CBF upon applying the Zero-CO_2_. Dorzolamide abolished the CBF decrease induced by applying the Zero-CO_2_. (**B**) Effects of brinzolamide (0.1 µM) on CBF upon applying the Zero-CO_2_. The addition of brinzolamide abolished the CBF decrease induced by the Zero-CO_2_. (**C**) Effects of brinzolamide on CBF changes induced by the NH_4_^+^ pulse. In the presence of brinzolamide, the application of the NH_4_^+^ pulse induced an immediate increase followed by a gradual increase in CBF. Cessation of the NH_4_^+^ pulse immediately decreased CBF. (**D**) Effects of brinzolamide on pH_i_ changes induced by the NH_4_^+^ pulse. Brinzolamide alone did not change the pH_i_. The subsequent application of the NH_4_^+^ pulse induced an immediate increase followed by a slight decrease in pH_i_. Cessation of the NH_4_^+^ pulse immediately decreased pH_i_, and then pH_i_ increased gradually to a control value. * shows control values.

**Figure 8 ijms-25-09069-f008:**
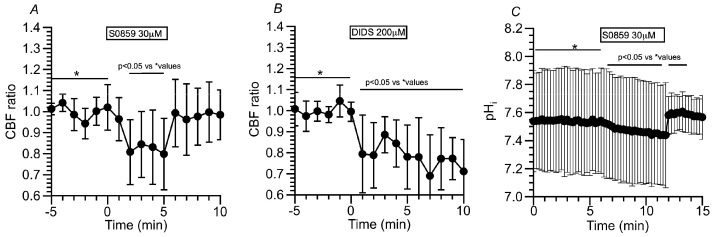
Effects of NBC inhibitors (S0859 and DIDS) on CBF and pH_i_ in c-hNECs. (**A**) S0859 (30 µM). The addition of S0859 decreased CBF. The removal of S0859 increased CBF to a control level. (**B**) DIDS (200 µM). The addition of DIDS decreased CBF. However, the removal of DIDS did not recover CBF. (**C**) Effects of S0859 on pH_i_. The addition of S0859 gradually decreased pH_i_, then the removal of S0859 increased pH_i_, and then the pH_i_ decreased gradually. * shows control values.

**Figure 9 ijms-25-09069-f009:**
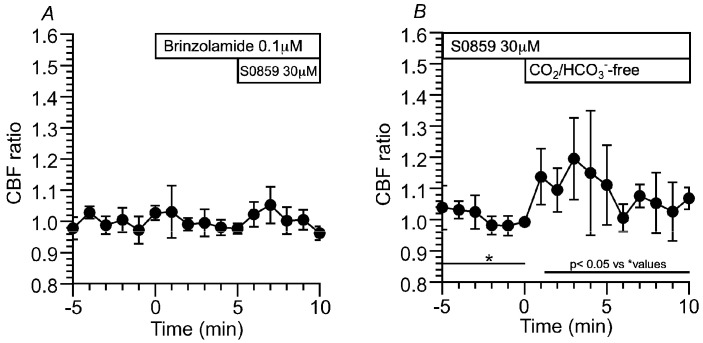
Effects of S0859 on CBF. (**A**) Prior treatment of brinzolamide. In the presence of brinzolamide, the addition of S0859 did not change CBF in c-hNECs. (**B**) The c-hNECs were treated with S0859 for 1 h. The application of Zero-CO_2_ transiently increased CBF in c-hNECs. * shows control values.

**Figure 10 ijms-25-09069-f010:**
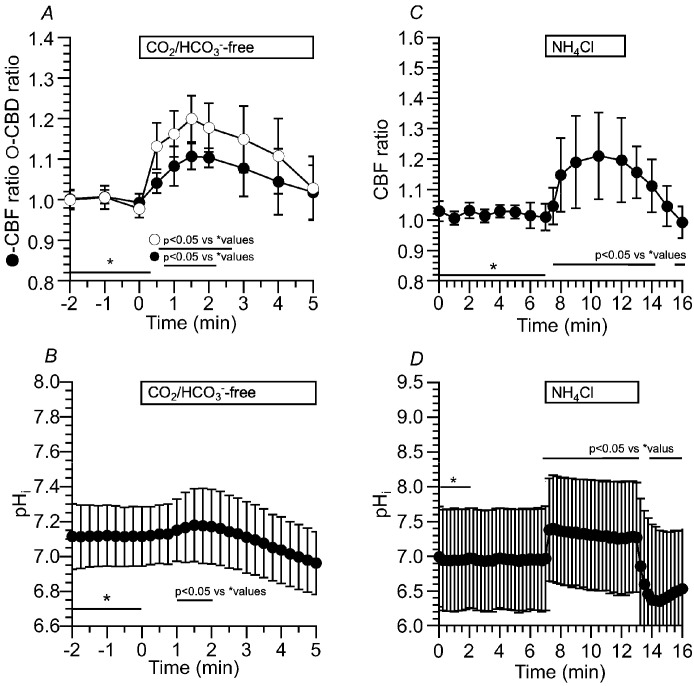
Effects of Zero-CO_2_ and the NH_4_^+^ pulse on CBF, CBD and pH_i_ in c-hBECs. (**A**) Application of Zero-CO_2_ transiently increased CBF and CBD in c-hBECs. (**B**) Application of Zero-CO_2_ transiently increased pH_i_ in c-hBECs. (**C**) Application of the NH_4_^+^ pulse in the control solution increased and plateaued CBF. Cessation of the NH_4_^+^ pulse decreased CBF. (**D**) Application of the NH_4_^+^ pulse immediately increased and plateaued pH_i_. Cessation of the NH_4_^+^ pulse decreased pH_i_ and then gradually increased to a control level. * shows control values.

**Figure 11 ijms-25-09069-f011:**
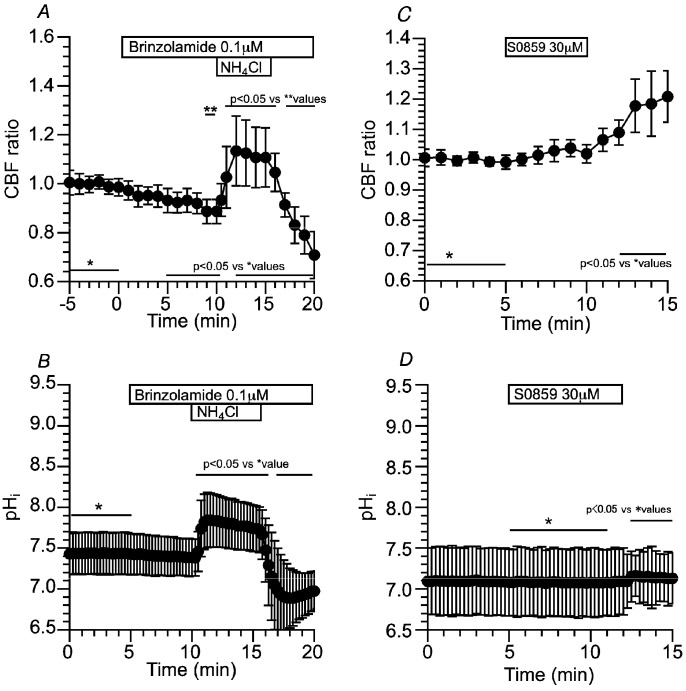
Effects of brinzolamide and S0859 on CBF and pH_i_ in c-hBECs. (**A**) The addition of brinzolamide gradually decreased CBF, and then application of the NH_4_^+^ pulse increased and plateaued CBF. Cessation of the NH_4_^+^ pulse decreased CBF. (**B)** The addition of brinzolamide (0.1 µM) gradually decreased pH_i_, and then the NH_4_^+^ pulse increased and plateaued pH_i_. Cessation of the NH_4_^+^ pulse decreased pH_i_. (**C**) The addition of S0859 slightly increased CBF (not significant). Removing S0859 gradually increased CBF. (**D**) The addition of S0859 did not change pH_i_. Removing S0859 slightly increased pH_i_. * shows control values.

**Figure 12 ijms-25-09069-f012:**
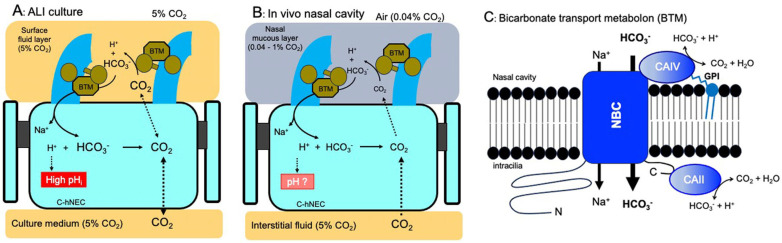
Schematic diagrams of CBF and pH_i_ regulation by CAIV in c-hNEC (**A**): ALI culture. CO_2_ (5%) supplied from the culture gas is converted to HCO_3_^−^ and H^+^ by CAIV. The HCO_3_^−^ transport metabolon (BTM, CAIV, NBC and CAII) maximizes the rate of HCO_3_^−^ entry in c-hNECs cultured by the ALI. The HCO_3_^−^ entered traps H^+^ and is converted to CO_2_ by CAII leading to an extremely high pH_i_ in c-hNECs. (**B**): In vivo. CO_2_, which is leaked from the interstitial fluid (5%) to the apical surface (0.04%), is converted to H^+^ and HCO_3_^−^ by CAIV. H^+^ stays in the surface mucous layer. The HCO_3_^−^ enters the cell via BTM. HCO_3_^−^, which is removed by CAII, enters and keeps an adequate pH_i_ and CBF in c-hNECs exposed to air and exhalation (0.04–1% CO_2_). At present, we do not know the exact pH_i_ of c-hNECs in vivo. (**C**): Bicarbonate transport metabolon (BTM). Schematic model of the physical and functional interactions of extracellular CAIV and cytoplasmic CAII with NBC in nasal cilia, based on NBC1 and CAIV co-transfected HEK293 cells [7]. CAIV is anchored to the apical surface via a glycosyl phosphatidyl inositol (GPI) anchor. CAIV converts CO_2_ to HCO_3_^−^ to increase a local high HCO_3_^−^ concentration of extraciliary fluid, which accelerates HCO_3_^−^ transport into the cilia. At the intraciliary surface, CAII immediately converts HCO_3_^−^ to CO_2_, which decreases the local intracellular HCO_3_^−^ concentration. Thus, BTM generates the gradient of HCO_3_^−^ concentration across the ciliary membrane, which increases the driving force for HCO_3_^−^ entry. The BTM drives HCO_3_^−^ transport into the cilia.

**Table 1 ijms-25-09069-t001:** Primers used to amplify CA.

Transcript	Direction	Sequence	Size (bp)
CAI	Sense	AGCTGCCTCAAAGGCTGATG	181
	Antisense	GGTCCAGAAATCCAGGGATGAA	
CAII	Sense	TTACTGGACCTACCCAGGCTCAC	167
	Antisense	GCCAGTTGTCCACCATCAGTTC	
CAIII	Sense	CATGAGAATGGCGACTTCCAGA	141
	Antisense	GAATGAGCCCTGGTAGGTCCAGTA	
CAIV	Sense	TCCCTAGAAACCTAGGGTCATTTCA	156
	Antisense	TGGAGCTAGATCACGTTTCACAA	
CAVb	Sense	TGTTCTGAAGTGAAAGTCTGGTCTG	172
	Antisense	CCAAACTAGAGTGCCCTGGATG	
CAVI	Sense	CTGTACTGGCAGCCTTCGTTGA	148
	Antisense	AGGTTCCTGGGCAGCATGTC	
CAVII	Sense	CAATGGCCACTCTGTCCAGGTA	102
	Antisense	AAGTGAAACTGCTTGAGGCGGTA	
CAIX	Sense	ACCAGACAGTGATGCTGAGTGCTAA	84
	Antisense	TCAGCTGTAGCCGAGAGTCACC	
CAXII	Sense	TGTACTGCACACACATGGACGAC	128
	Antisense	TCCTGCCGCAGTACAGACTTG	
CAXIV	Sense	TGTAGGAATCTTGGTTGGCTGTCTC	134
	Antisense	TTTATGCCTCAGTCGTGGCTTG	
GAPDH	Sense	GCACCGTCAAGGCTGAGAAC	180
	Antisense	TGGTGAAGACGCCAGTGGA	

**Table 2 ijms-25-09069-t002:** Primers used to amplify NBCs and AEs.

NBC			
Transcript	Direction	Sequence	Size (bp)
NCBE (SLC4A10)	Sense	GCAGGTCAGGTTGTTTCTCCTC	498
	Antisense	TCTTCCTCTTCTCCTGGGAAGG	
NBC1 (SLC4A11)	Sense	GGCCTGTGGAACAGTTTCTTCC	690
	Antisense	TGCCCTTCACCAGCCTGTTCTC	
NBCe1 (SLC4A4)	Sense	GGTGTGCAGTTCATGGATCGTC	336
	Antisense	GTCACTGTCCAGACTTCCCTTC	
NBCe2 (SLC4A5)	Sense	ATCTTCATGGACCAGCAGATCAC	468
	Antisense	TGCTTGGCTGGCATCAGGAGG	
NBCn1 (SLC4A7)	Sense	CAGATGCAAGCAGCCTTGTGTG	328
	Antisense	GGTCCATGATGACCACAAGCTG	
NDCBE1 (SLC4A8)	Sense	GCTCAAGAAAGGCTGTGGCTAC	243
	Antisense	CATGAAGACTGAGCAGCCCATC	
**AE**			
AE2	Sense	GAAGATTCCTGAGAATGCCT	181
	Antisense	GTCCATGTTGGCAGTAGTCG	
AE3	Sense	ATCTGAGGCAGAACCTGTGG	418
	Antisense	TTTCACTAAGTGTCGCCGC	
SLC26A4	Sense	GTTTACTAGCTGGCCTTATATTTGGACTGT	484
	Antisense	AGGCTATGGATTGGCACTTTGGGAACG	
SLC26A6	Sense	TAGGGGAGGTTGGGCCAGGGATGC	456
	Antisense	TGCCGGGAAGTGCCAAACAGGAAGAAGTAGAT	
SLC26A9	Sense	TCCAGGTCTTCAACAATGCCAC	400
	Antisense	CGAGTCTTGTGCATGTAGCGAG	

## Data Availability

The raw data supporting the conclusions of this article will be made available by the authors on request.

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
