# Peer review of "Ciliary Motility Decreased by a CO2/HCO3-Free Solution in Ciliated Human Nasal Epithelial Cells Having a pH Elevated by Carbonic Anhydrase IV"

_ijms, 2024, doi:10.3390/ijms25169069_

Round 1

Reviewer 1 Report (New Reviewer)

Comments and Suggestions for Authors

The study has been well designed and the methods of experiments are described appropriately.  However, some of the results are not presented clearly.

1. The figure legend of Figure 6 (C) states that the application of NH4+ pulse induced an immediate increase and cessation of the NH4+ pulse immediately decreased the pHi.  In the figure 6(C), the bar below "p<0.05 vs *values" ranges from the start of application to the end of cessation of the NH4+.  If both the increase and decrease are statistically significant, this bar should be interrupted as in the Figure 7(D).

2.  Figure 8 (C) and its legend are also as unclear as Figure 6 (C).

Author Response

THE STUDY HAS BEEN WELL DESIGNED AND THE METHODS OF EXPERIMENTS ARE DESCRIBED APPROPRIATELY.  HOWEVER, SOME OF THE RESULTS ARE NOT PRESENTED CLEARLY.

  1. THE FIGURE LEGEND OF FIGURE 6 (C) STATES THAT THE APPLICATION OF NH4+ PULSE INDUCED AN IMMEDIATE INCREASE AND CESSATION OF THE NH4+ PULSE IMMEDIATELY DECREASED THE PHI.  IN THE FIGURE 6(C), THE BAR BELOW "P<0.05 VS *VALUES" RANGES FROM THE START OF APPLICATION TO THE END OF CESSATION OF THE NH4+.  IF BOTH THE INCREASE AND DECREASE ARE STATISTICALLY SIGNIFICANT, THIS BAR SHOULD BE INTERRUPTED AS IN THE FIGURE 7(D).

I corrected the Fig. 7D according to the reviewer’s comments.

  1. FIGURE 8 (C) AND ITS LEGEND ARE ALSO AS UNCLEAR AS FIGURE 6 (C).

I corrected the Fig. 8C according to the reviewer’s comments

Reviewer 2 Report (New Reviewer)

Comments and Suggestions for Authors

The manuscript entitled:” A decrease in ciliary motility induced by a CO2/HCO3-free solution characteristic of ciliated human nasal epithelial cells: acceleration of HCO3- transport by carbonic anhydrase IV” by Shota Okamoto and co-Authors describes changes in the intracellular pH (pHi) in human nasal and bronchial cells grown under different conditions (Zero-CO2, NH4+, CA inhibitors, NBC inhibitors) and how such changes affect cilia beating (frequency and amplitude).

Generally, although the presented data might be interesting, the manuscript is poorly written and thus difficult to follow. It requires extensive re-writing and should be corrected by an English native speaker with knowledge of cell biology.  

The Introduction did not provide sufficient background information and thus the manuscript will be hardly understandable for the reader outside the field unfamiliar with the regulation of the intracellular pH. In some places, it is a mixture of the randomly provided information regarding nasal epithelial cells, previous results, and results presented in this manuscript. Thus it is very hard to follow.

I would suggest starting with the definition of intracellular pH, and a short description/listing of the factors affecting pHi. Next, I would suggest focusing on HCO3- and CO2 + H2O ↔  + H+  reaction and clearly describe them mentioning also that Carbonic anhydrase (CA) is an enzyme that catalyzes the hydration and dehydration of carbon dioxide. It would be also helpful to add where are HCO3- ions, how they are transported across the membrane, how HCO3- affects the level of H+ (why it causes reduction of H+ and increase of the pHi) and how CO2 affects the level of H+.  

Next, The Authors could introduce cells of interest and provide information about the level of CO2, HCO3- in the sinuses and trachea/lung and describe the scientific problem they intend to address in this manuscript, stating what is already known and what they found.

lines 56-61 - what is the difference between this and previous studies [6]. Why there are different effects on CBF under Zero-CO2 conditions.

The Results section is also confusing. The reader can only assume why the experiments are performed. The titles of the results sections are not informative (e.g. 2.1.1.”  PCR” - such a title informs about the method used not about the outcome of the performed analyses).

Suggested title of the Results section: 2.1. Carbonic anhydrase CAIV is expressed in c-hNEC cells but not in c-hBEC cells. Please change the titles of other chapters accordingly.

Also, it would be helpful if the Authors provided the description with a scientific question to be answered or put a hypothesis to be addressed at the beginning of each subchapter (to highlight the goal of the undertaken research).

The description of CA expression can be shortened. Such shortening will also improve the description clarity.

Compared to other CAs, the level of CAIV is low (Fig 1). I wonder if the primers are optimal. Did the Authors test these primers on mRNA isolated from cells known to highly expressed CAIV? Why CAIV having such a limited expression compared to other CAs would play an important role in hNEC cells?

Why the Authors also examined the levels of CAs and NBCs or AEs in cells from biopsies? What is the conclusion?

WB of CAIV – what is the theoretical molecular weight of this enzyme based on its amino acid sequence? The specific band of AF21816 (https://www.rndsystems.com/products/human-carbonic-anhydrase-iv-ca4-antibody_af2186) is approximately 30 kDa (between 25 and 37 kDa markers) not as the Authors stated 33-40 kDa). Please add that PNGaseF is a recombinant glycosidase.

The IF images, especially acetylated tubulin staining, are of poor quality. Based on the DAPI staining it seems that the nuclei are affected by the fixation method. Much nicer images were provided in the preprint published on Research Square (in the preprint the low level of CAIV was detected in hBEC cilia). Some information in 2.1.3. subchapter belongs to the Fig3 description.

Expression of NBC and AE – I would suggest shortening this subchapter by stating that the expression of the following subtypes of NBCs and AEs were analyzed in hNEC and hBEC cells but no differences between those cells were detected.

CBF and CBD (lines 166-170).  The Authors stated that:”

In c-hNECs, …the ciliary bend distance (CBD) was 54.0 ± 27.2 μm (n = 11). In c-hBECs, …he CBD was 65.3 ± 8.0 μm (n = 12). There was no difference in … or CBD between c-hNECs and c-hBECs.”

The SD in the case of hNEC cells is rather large (half of the average value). This indicates a substantial variation in the CBD values (why?). In such a case I would recommend providing a graph showing all measurements.

Lines 183-184: The application of Zero-CO2 causes a decrease in the pHi (I agree with this). The Authors explain that such a decrease is due to H+ production. Is this an H+ production or reduced H+ elimination due to the low level of HCO3-?

Lines 190-191 – unintended repetition?

Generally, while describing the effect of Zero-CO2, NH4+, or drugs on CBF and CBD, I would suggest first describing the effect of those agents on pHi and next that such a change in pHi translates into altered CBF, CBD. The changes in the CBF and CBD are only phenotypic manifestations of the changes in the pHi. Describing in this way will help the reader to follow the presented data.

2.3.1.3. Please whenever possible, describe together the effect (or its lack) of the applied inhibitors on cilia beating to avoid unnecessary repetitions.   

Line 249: isn’t CAIV a membrane-associated enzyme? (“Carbonic anhydrase (CA) IV is expressed on the extracellular surfaces of endothelial and epithelial cells and attached by a glycosylphosphatidylinositol anchor rather than a membrane-spanning domain.” From https://doi.org/10.1016/B978-0-444-63258-6.00006-8

Line 250: The Authors stated that:” CAIV is suggested to interact with CAII.” – is this statement made based on the Authors' data presented in this manuscript or based on previous data (REF?)?

2.3.1.4 - Please whenever possible, describe together the effect (or its lack) of the applied inhibitors on cilia beating to avoid unnecessary repetitions. Are the reported differences in CBF statistically significant?

Line 293:” These results suggest that the interactions of CAII, CAIV and NBC accelerate….”

The Authors did not show that those proteins interact (e.g. using pull-down assay).

Those proteins may simply act in the subsequent/different steps of the same process.

Comments on the Quality of English Language

the manuscript is poorly written and thus difficult to follow. It requires extensive re-writing and should be corrected by an English native speaker with knowledge of cell biology. 

Author Response

THE MANUSCRIPT ENTITLED:” A DECREASE IN CILIARY MOTILITY INDUCED BY A CO2/HCO3-FREE SOLUTION CHARACTERISTIC OF CILIATED HUMAN NASAL EPITHELIAL CELLS: ACCELERATION OF HCO3- TRANSPORT BY CARBONIC ANHYDRASE IV” BY SHOTA OKAMOTO AND CO-AUTHORS DESCRIBES CHANGES IN THE INTRACELLULAR PH (PHI) IN HUMAN NASAL AND BRONCHIAL CELLS GROWN UNDER DIFFERENT CONDITIONS (ZERO-CO2, NH4+, CA INHIBITORS, NBC INHIBITORS) AND HOW SUCH CHANGES AFFECT CILIA BEATING (FREQUENCY AND AMPLITUDE). 

GENERALLY, ALTHOUGH THE PRESENTED DATA MIGHT BE INTERESTING, THE MANUSCRIPT IS POORLY WRITTEN AND THUS DIFFICULT TO FOLLOW. IT REQUIRES EXTENSIVE RE-WRITING AND SHOULD BE CORRECTED BY AN ENGLISH NATIVE SPEAKER WITH KNOWLEDGE OF CELL BIOLOGY.

I have carefully checked English.  And we sent the manuscript to the IJMS English check before the submission of the revised manuscript.

THE INTRODUCTION DID NOT PROVIDE SUFFICIENT BACKGROUND INFORMATION AND THUS THE MANUSCRIPT WILL BE HARDLY UNDERSTANDABLE FOR THE READER OUTSIDE THE FIELD UNFAMILIAR WITH THE REGULATION OF THE INTRACELLULAR PH. IN SOME PLACES, IT IS A MIXTURE OF THE RANDOMLY PROVIDED INFORMATION REGARDING NASAL EPITHELIAL CELLS, PREVIOUS RESULTS, AND RESULTS PRESENTED IN THIS MANUSCRIPT. THUS IT IS VERY HARD TO FOLLOW. 

I WOULD SUGGEST STARTING WITH THE DEFINITION OF INTRACELLULAR PH, AND A SHORT DESCRIPTION/LISTING OF THE FACTORS AFFECTING PHI. NEXT, I WOULD SUGGEST FOCUSING ON HCO3- AND CO2 + H2O ↔  + H+  REACTION AND CLEARLY DESCRIBE THEM MENTIONING ALSO THAT CARBONIC ANHYDRASE (CA) IS AN ENZYME THAT CATALYZES THE HYDRATION AND DEHYDRATION OF CARBON DIOXIDE. IT WOULD BE ALSO HELPFUL TO ADD WHERE ARE HCO3- IONS, HOW THEY ARE TRANSPORTED ACROSS THE MEMBRANE, HOW HCO3- AFFECTS THE LEVEL OF H+ (WHY IT CAUSES REDUCTION OF H+ AND INCREASE OF THE PHI) AND HOW CO2 AFFECTS THE LEVEL OF H+.  

NEXT, THE AUTHORS COULD INTRODUCE CELLS OF INTEREST AND PROVIDE INFORMATION ABOUT THE LEVEL OF CO2, HCO3- IN THE SINUSES AND TRACHEA/LUNG AND DESCRIBE THE SCIENTIFIC PROBLEM THEY INTEND TO ADDRESS IN THIS MANUSCRIPT, STATING WHAT IS ALREADY KNOWN AND WHAT THEY FOUND. 

I have rewritten the introduction according to the reviewer’s suggestion.

Please see “introduction”.

LINES 56-61 - WHAT IS THE DIFFERENCE BETWEEN THIS AND PREVIOUS STUDIES [6]. WHY THERE ARE DIFFERENT EFFECTS ON CBF UNDER ZERO-CO2 CONDITIONS. 

In the previous study, we did not understand what induces a small transient increase upon applying the Zero-CO2. The present study revealed that the high pHi causes a decrease in pHi upon applying the Zero-CO2. In the previous study, cells were kept in an cooled solution (4°C) to avoid cellular damages. In this experiments, we keep cells at the room temperature to keep the ion transport. The different responses in pHi and CBF upon applying the Zero-CO2 is caused by the cellular condition. In cells kept at the cooled solution, pHi decreased approximately 0.1 unit. We added these in Discussion as follows.

P.12 5th paragraph

“The application of Zero-CO2 induced various responses in CBF and pHi, decreases (Fig. 5) or small increases (Fig. 6A) [6], although it never induced large increases as shown in c-hBECs (Fig. 10) or bronchial ciliated cells [1]. The responses were affected by cellular conditions used for the experiments. In this study, c-hNECs with a permeable support filter were kept in the control solution at room temperature until the measurement of CBF or pHi,. The pHi leading to changes in CBF depends on the conditions of c-hNECs kept until experiments, such as time and temperature. Keeping cells at 4° C in air more than 3 hrs, the pHi decreased by approximately 0.1-0.15, and in the c-hNECs with a decreased pHi, the application of Zero-CO2 induced no changes or small increases in pHi and CBF.”

THE RESULTS SECTION IS ALSO CONFUSING. THE READER CAN ONLY ASSUME WHY THE EXPERIMENTS ARE PERFORMED. THE TITLES OF THE RESULTS SECTIONS ARE NOT INFORMATIVE (E.G. 2.1.1.”  PCR” - SUCH A TITLE INFORMS ABOUT THE METHOD USED NOT ABOUT THE OUTCOME OF THE PERFORMED ANALYSES). 

SUGGESTED TITLE OF THE RESULTS SECTION: 2.1. CARBONIC ANHYDRASE CAIV IS EXPRESSED IN C-HNEC CELLS BUT NOT IN C-HBEC CELLS. PLEASE CHANGE THE TITLES OF OTHER CHAPTERS ACCORDINGLY.

 I have changed the title of the section.

ALSO, IT WOULD BE HELPFUL IF THE AUTHORS PROVIDED THE DESCRIPTION WITH A SCIENTIFIC QUESTION TO BE ANSWERED OR PUT A HYPOTHESIS TO BE ADDRESSED AT THE BEGINNING OF EACH SUBCHAPTER (TO HIGHLIGHT THE GOAL OF THE UNDERTAKEN RESEARCH). 

I have added the description for the readers according to the reviewer’s comments

THE DESCRIPTION OF CA EXPRESSION CAN BE SHORTENED. SUCH SHORTENING WILL ALSO IMPROVE THE DESCRIPTION CLARITY. 

COMPARED TO OTHER CAS, THE LEVEL OF CAIV IS LOW (FIG 1). I WONDER IF THE PRIMERS ARE OPTIMAL. DID THE AUTHORS TEST THESE PRIMERS ON MRNA ISOLATED FROM CELLS KNOWN TO HIGHLY EXPRESSED CAIV? WHY CAIV HAVING SUCH A LIMITED EXPRESSION COMPARED TO OTHER CAS WOULD PLAY AN IMPORTANT ROLE IN HNEC CELLS?

Previous study demonstrated that nasal epithelia express 11 CAs (Kim et al., Laryngoscope 2008, 118, 1856). CAII, IV, Vb mRNAs are highly expressed in nasal mucosa. Therefore we choosed CA I, II. III, IV, V in this study. Fortunately we found the difference in CAIV expression between c-hNECs and c-hBECs. The contribution of the other CAs is unknown.

WHY THE AUTHORS ALSO EXAMINED THE LEVELS OF CAS AND NBCS OR AES IN CELLS FROM BIOPSIES? WHAT IS THE CONCLUSION?

 The previous studies performed in the native nasal tissue. However, they did not examine in the cultured cells. The experiments for measuring CBF, CBD and pHi were carried out using the cultured cells. Therefore, we examined the expression using the cultured cells.

WB OF CAIV – WHAT IS THE THEORETICAL MOLECULAR WEIGHT OF THIS ENZYME BASED ON ITS AMINO ACID SEQUENCE? THE SPECIFIC BAND OF AF21816 (HTTPS://WWW.RNDSYSTEMS.COM/PRODUCTS/HUMAN-CARBONIC-ANHYDRASE-IV-CA4-ANTIBODY_AF2186) IS APPROXIMATELY 30 KDA (BETWEEN 25 AND 37 KDA MARKERS) NOT AS THE AUTHORS STATED 33-40 KDA). PLEASE ADD THAT PNGASEF IS A RECOMBINANT GLYCOSIDASE. 

The data sheet provides the 33 kDa. The data sheet also provide the CAIV band obtained by the simple WD (40 kDa). We did mistake in the reading of the data sheet.

In the previous experiments using MAB2186, we detected the band of CAIV at 47 kDa. In this study, the treatment of PNGaseF decreased to 40 kDa. We need another treatment to decrease the molecular mass of CAIV. We think the band detected by aF2186 show CAIV, although the molecular mass was still larger than 33 kDa..

THE IF IMAGES, ESPECIALLY ACETYLATED TUBULIN STAINING, ARE OF POOR QUALITY. BASED ON THE DAPI STAINING IT SEEMS THAT THE NUCLEI ARE AFFECTED BY THE FIXATION METHOD. MUCH NICER IMAGES WERE PROVIDED IN THE PREPRINT PUBLISHED ON RESEARCH SQUARE (IN THE PREPRINT THE LOW LEVEL OF CAIV WAS DETECTED IN HBEC CILIA). SOME INFORMATION IN 2.1.3. SUBCHAPTER BELONGS TO THE FIG3 DESCRIPTION.

I added the new IF figures as Fig. 3.

EXPRESSION OF NBC AND AE – I WOULD SUGGEST SHORTENING THIS SUBCHAPTER BY STATING THAT THE EXPRESSION OF THE FOLLOWING SUBTYPES OF NBCS AND AES WERE ANALYZED IN HNEC AND HBEC CELLS BUT NO DIFFERENCES BETWEEN THOSE CELLS WERE DETECTED. 

 We shortened the description in section 2.2.

CBF AND CBD (LINES 166-170).  THE AUTHORS STATED THAT:”

IN C-HNECS, …THE CILIARY BEND DISTANCE (CBD) WAS We need another treatment to decrease the molecular mass of CAIV.. IN C-HBECS, …HE CBD WAS 65.3 ± 8.0 ΜM (N = 12). THERE WAS NO DIFFERENCE IN … OR CBD BETWEEN C-HNECS AND C-HBECS.”

THE SD IN THE CASE OF HNEC CELLS IS RATHER LARGE (HALF OF THE AVERAGE VALUE). THIS INDICATES A SUBSTANTIAL VARIATION IN THE CBD VALUES (WHY?). IN SUCH A CASE I WOULD RECOMMEND PROVIDING A GRAPH SHOWING ALL MEASUREMENTS. 

I added the results. The CBD of c-hNECs were 62.2 ± 13.8 Μm (n=20).

Please see p.5 section 2.3.

LINES 183-184: THE APPLICATION OF ZERO-CO2 CAUSES A DECREASE IN THE PHI (I AGREE WITH THIS). THE AUTHORS EXPLAIN THAT SUCH A DECREASE IS DUE TO H+ PRODUCTION. IS THIS AN H+ PRODUCTION OR REDUCED H+ ELIMINATION DUE TO THE LOW LEVEL OF HCO3-?

Thank you for the suggestion. I added “H+ production and no H+ elimination” according to the reviewer’s comments

LINES 190-191 – UNINTENDED REPETITION?

I removed the repetition.

GENERALLY, WHILE DESCRIBING THE EFFECT OF ZERO-CO2, NH4+, OR DRUGS ON CBF AND CBD, I WOULD SUGGEST FIRST DESCRIBING THE EFFECT OF THOSE AGENTS ON PHI AND NEXT THAT SUCH A CHANGE IN PHI TRANSLATES INTO ALTERED CBF, CBD. THE CHANGES IN THE CBF AND CBD ARE ONLY PHENOTYPIC MANIFESTATIONS OF THE CHANGES IN THE PHI. DESCRIBING IN THIS WAY WILL HELP THE READER TO FOLLOW THE PRESENTED DATA. 

I corrected the description according to the reviewer’s comments

2.3.1.3. PLEASE WHENEVER POSSIBLE, DESCRIBE TOGETHER THE EFFECT (OR ITS LACK) OF THE APPLIED INHIBITORS ON CILIA BEATING TO AVOID UNNECESSARY REPETITIONS.   

I corrected the description according to the reviewer’s comments

LINE 249: ISN’T CAIV A MEMBRANE-ASSOCIATED ENZYME? (“CARBONIC ANHYDRASE (CA) IV IS EXPRESSED ON THE EXTRACELLULAR SURFACES OF ENDOTHELIAL AND EPITHELIAL CELLS AND ATTACHED BY A GLYCOSYLPHOSPHATIDYLINOSITOL ANCHOR RATHER THAN A MEMBRANE-SPANNING DOMAIN.” FROM HTTPS://DOI.ORG/10.1016/B978-0-444-63258-6.00006-8

I corrected the description according to the reviewer’s comments.

LINE 250: THE AUTHORS STATED THAT:” CAIV IS SUGGESTED TO INTERACT WITH CAII.” – IS THIS STATEMENT MADE BASED ON THE AUTHORS' DATA PRESENTED IN THIS MANUSCRIPT OR BASED ON PREVIOUS DATA (REF?)?

We do not confirm the metabolon in c-hNECs. We toned down the description. 

2.3.1.4 - PLEASE WHENEVER POSSIBLE, DESCRIBE TOGETHER THE EFFECT (OR ITS LACK) OF THE APPLIED INHIBITORS ON CILIA BEATING TO AVOID UNNECESSARY REPETITIONS. ARE THE REPORTED DIFFERENCES IN CBF STATISTICALLY SIGNIFICANT?

We removed the repetition.

Changes in CBF were significant (detected by 1 way ANOVA). 

LINE 293:” THESE RESULTS SUGGEST THAT THE INTERACTIONS OF CAII, CAIV AND NBC ACCELERATE….” THE AUTHORS DID NOT SHOW THAT THOSE PROTEINS INTERACT (E.G. USING PULL-DOWN ASSAY). THOSE PROTEINS MAY SIMPLY ACT IN THE SUBSEQUENT/DIFFERENT STEPS OF THE SAME PROCESS. 

 We do not detect the metabolon in c-HNECs. I changed the description.

Comments on the Quality of English Language

THE MANUSCRIPT IS POORLY WRITTEN AND THUS DIFFICULT TO FOLLOW. IT REQUIRES EXTENSIVE RE-WRITING AND SHOULD BE CORRECTED BY AN ENGLISH NATIVE SPEAKER WITH KNOWLEDGE OF CELL BIOLOGY. 

I carefully rewrote the manuscript.

Round 2

Reviewer 2 Report (New Reviewer)

Comments and Suggestions for Authors

ijms-3100747- corrected version

A decrease in ciliary motility induced by a CO2/HCO3--free solution characteristic of ciliated human nasal epithelial cells: acceleration of HCO3- transport by carbonic anhydrase IV

The clarity of the manuscript was partly improved however, I still found some problems (see below).  Some parts of the discussion can be transferred to the introduction to provide a better background for the reader.

Major concern.

1.       The Authors stated that “The goal of this study is to clarify the CAIV-mediated mechanism, which suppresses increases in CBF, CBD and pHi in c-hNECs upon applying the Zero-CO2. Is it possible that other membrane-bound CAs (CAIX) could take over CAIV functions and thus the relation between the observed phenotype and CAIV is not that straightforward. CA IX was detected in the cytoplasm and plasma membrane of epithelial cells (Kim et al., 2008).

2.       The Authors used two CA inhibitors, brinzolamide (inhibits CA II) and dorzolamide (inhibits CA II and CAIV).  They found that “brinzolamide and dorzolamide similarly inhibit the Eq.1 in c-hNECs….” and hypothesized that therefore “The similar effects of brinzolamide and dorzolamide on CBF, CAIV may interact with CAII in c-hNECS, as previously reported in the bicarbonate transport metabolon [7, 8, 12].” The identical/ similar effect of the inhibition of CAII and CAII + CAIV may also suggest that CII is responsible for the observed alterations, especially that CAIV is expressed at the very low level. The synthesis of the specific CAIV inhibitors was recently reported (Carbonic Anhydrase IV Selective Inhibitors Counteract the Development of Colitis-Associated Visceral Pain in Rats. Lucarini E, Nocentini A, Bonardi A, Chiaramonte N, Parisio C, Micheli L, Toti A, Ferrara V, Carrino D, Pacini A, Romanelli MN, Supuran CT, Ghelardini C, Di Cesare Mannelli L. Cells. 2021 Sep 26;10(10):2540. doi: 10.3390/cells10102540. Is there an inhibitor that specifically inhibits CAIX?

3.       It is confusing why the same conditions could cause “decrease or a small transient increase of CBF (discussion, lines 364-365)

Minor issues:

Abstract: needs to be partly re-written (for clarity reason). Suggestion: start with the periodic exposure of Nasal epithelial cells to low CO2 (breathe in). Then ask a question how such a conditions affect cilia beating or how cells respond to such conditions. Summarize data and follow with the short conclusion.

Introduction

Line 42: - missing comma: ….AE), and Na+/H+ exchanger…

Line 43: spelling mistake: the cullualr metabolism,

Line 42-43: please add that CO2 is transported from the extracellular fluids via diffusion (correct?) and that its accumulation inside the cell, will cause cell acidification (High level of H+, low pHi). - is this correct?

Line 45: could you add H2O to the equitation?  (H2O + CO2…..

Lines 47 and 48: “CO2 entered” and „ HCO3- entered” – did Authors mean increase of the level of…CO2 or HCO3-? If yes, could you add how it happens (diffusion, product of metabolism – CO2), membrane exchanger, co-transporters (HCO3-) ?

Line 50: spelling mistake: “epitheliium 

Lines 50 and 51: “…The ciliated nasal epitheliium is an unique tissue placed under an unusual CO2 conditions”  - maybe the Authors could add (if the information is correct) that the level of CO2 cycles between 0.04% (breathe in) and 1% (breathe out)

Lines 58-60: “The periodic air exposure (0.04% CO2) appears to increase pHi leading to a CBF increase in the ciliated nasal epithelial cells, as shown in tracheal and lung airway ciliated cells upon applying the CO2/HCO3--free solution (Zero-CO2) [1, 4, 5].” – please provide this information earlier, just after “The ciliated nasal epitheliium is an unique tissue placed under an unusual CO2 condition: the apical surface is periodically exposed to the fresh air containing an extremely low concentration of CO2 (0.04%) with respiration.”

Lines 62-63: ” Ciliated human nasal epithelial cells (c-hNECs) were differentiated from human nasal epithelia (operation samples) by the air liquid interface (ALI) culture.” – as it is now this information belongs rather to the Methods section. Suggested change: “Ciliated human nasal epithelial cells (c-hNECs) are differentiated by the air liquid interface (ALI) culture.”

Lines 63 and next. Could you please combine information from [REF2] and next clearly highlight what is different in tracheal cilia [REF1]

Line 72: “A previous study showed that 11 CA subtypes are ex-…” - please for clarity describe the CAs in a new paragraph. While describing CAs, please add that CAII is cytoplasmic and CAIV is membrane bound. Please shift the following information from the discussion section to Introduction: “The C-terminal tail of CAIV is anchored in the outer surface of the plasma membrane and a physical interaction between extracellular CAIV and NBC1 occurs via fourth extracellular loop of NBC1 [7] and perhaps include a modified fig 12 showing CAII, CAIV, and cotransporters  localization. It will be very helpful for the reader outside the field to understand from the very beginning that CAIV activity changes the levels of CO2, H+, and HCO3- outside the cell, in the nasal mucous layer and these extracellular changes affect intracellular pHi and cilia beating (is this correct?)

Lines 67-68: ” Moreover, an application of NH4+ pulse induced gradual decreases in pHi and CBF following to their immediate increases, suggesting that H+ is produced during the NH4+ pulse [2]. – please explain why (for the reader outside the field) – does NH4+ competes with H+ to bind HCO3-? Does in happens inside the cell or in the mucous layer? The answers to these questions are likely obvious to the Authors but might be not to the reader with general cilia knowledge but unfamiliar with pHi regulation.

Line 73:” However, the role of CAIV is not fully understood in nasal epithelia,…”. The Authors stated in previous sentence that “…including CAII and CAIV”. The reader may not understand why the presence of CAII and CAIV in nasal epithelial cells was highlighted. If these two were not previously studied, why in the following sentence, Authors refer only to CAIV? If CAII was studied, please provide the information, if not say that CAII and CAIV were not investigated but here we focused on CAIV because…… Is the role of other CAs in nasal epithelial cells “fully understood” in contrast to CAIV? Or maybe the Authors selected CAIV as a continuation of the previous study? Claiming such a research continuation also can be a good explanation why these studies were undertaken.

Line 74:” Previous studies also…” – please remove “also”

Line 76:” in HEK293 cells transfected CAIV and HCO3- transporters” – suggestion “in co-transfected HEK293 cells”

Line 77: “Nasal epithelia have also been shown…” – suggestion ”Moreover, it was shown that NBC [6] and AE [11] are expressed in nasal epithelia”

Line 78:” In c-hNECs, CAIV…” – suggestion, “Thus, In c-hNECs, CAIV…”

Results

Line 92:” 2.1. “CAIV expression in c-hNECs and not in c-hBEcs” – suggestion:” CAIV expression in c-hNECs but not in c-hBEcs”

I would also suggest to remove subchapters titles (2.1.1, 2.1.2, 2.1.3) and simply start description of detection by IF or WB as a new paragraph.

Line 94: suggest to change:  “The mRNA of CAIV…” – suggested change: “Interestingly, in contrast to other analyzed CAs, the mRNA of CAIV….”….The analyses of the expression levels of CAIV by Real-Time PCR revealed that mRNA were significantly higher in c-hNECs than in c-hBECs.

Lines 101-103: the Authors describe level of CAs in samples from patients. This fragment need an introduction. E.g. we were wondering if the level of CAs is altered in patients suffering from chronic sinusitis (why these conditions were selected?). What is different (in mucus layer) between healthy individuals and patients suffering chronic sinusitis?

Line 110- please remove the subtitle. State: “The western blot analyses are consisted with Real-Time PCR data. We used two anti-CAIV antibodies (MAB2186 and AF2186, R&D System).”

Line 127: suggested change” a single band detected was decreased to 40 kDa, but was still larger than 33 kD. Thus, the treatment with PNGaseF did not completely brake the glycosylation sites. Another treatment may be required…”

Line 133:” 2.1.3. Immunofluorescence examination for CAIV – please, remove.

This paragraph can be further shortened.

For example:” Immunofluorescence examinations for CAIV were carried out in c-hNECs (Fig. 3A) and c-hBECs (Fig. 3B) using AF2186, anti-CAIV antibodies and anti-acetylated tubulin antibodies (a cilia marker). The double-labelling analyses  clearly showed that cilia present in the apical surface of the c-hNECs (Fig. 3A2) are positively stained for both CAIV and AC-tubulin while in c-hBECs cells cilia the CAIV was detected (Fig. 3B1).

(information regarding DAPI staining can be only in the Fig description as this is general staining of the nuclei).

Line 150:” 2.2. Expression of NBC and AE mRNAs detected by RT-PCR in c-hNECs and c-hBECs”

Line 153: spelling “exprtessed”  „thet

„Previous studies have already shown thet NBC and AE are expressed in hunan nasal epithelia [15, 16, 17] and mice bronchiolar epithelia 154 [18].

Suggestion to change:” …that agrees with the previous studies

in hunan (please correct spelling) nasal epithelia [15, 16, 17] and mice bronchiolar epithelia 154 [18].

Line 155 ” ….NDCBE1 and AE2…” – please insert comma “NDCBE1, and AE2”

Line 163: please add introductory sentence. For example. First we characterized ciliary beating in HNEC and hBEC cells (follow with description in lines 163-167).

Next we analyze how low level of CO2 affect cilia beating. Under (what kind of conditions?) In c-hNECs, CBF was 8.39 ± 2.44 Hz (n = 47) and CBD was 62.2 ± 13.8 μm (n = 20) in 163 c-hNECs.

Line 166: suggestion: Thus, there was no…(please add “Thus” as this is a conclusion)

Line line 170: suggestion (need introduction” Next, in order to understand….we analyzed how minimal CO2 concentration typical for the air, affect ciia beating and pHi.

Line 176: the Authors wrote : “The pHi of c-hNECs in the control solution was 7.64 ± 0.19 (n = 9).” But in the line 213 is:” 6C). The pHi was 7.55 in c-hNECs perfused with the control solution.”  In both cases the Authors mean the same “control solution” if not, can be these solution re-named to avoid confusion? If this is the same solution, why pHi is different?

Line 181:” because of no HCO3- entry keeping H+ supply from other systems” – please clarify – because of the low level of CO2 (0.04%) or HCO3- (but inside the cell – so low production due to low CO2? What about the transport of HCO3- by co-exchanger and co-transporters?

Line 183:” caused by the different condition of cells used for the experiments.” – could you be more precise regarding differences in the conditions?

Lines 199-200:” We applied the NH4+ pulse in c-hNECs, upon applying the Zero-CO2. The switch to

the Zero-CO2 slightly increased CBF, but the increase in CBF was not significant” – it seems that the same information was repeated in this sentence.

Lines 253-254: “In the presence of brinzolamide, the NH4+ pulse immediately increased CBF

anf then gradually increased it without any CBF decrease.” – I do not understand this part “anf then gradually increased it without any CBF decrease.” Is this an increased followed by a decrease?

Line 256:’ In the presence of brinzolamide, the NH4+ pulse increased CBF without any CBF decrease in c-hNECs” – please correct. I do not understand “the increase in cilia beat frequency without any decrease of cilia beat frequency”.

Lines 267-269:” The similar effects of brinzolamide and dorzolamide on CBF, may suggests that (???) CAIV 267 may interact with CAII in c-hNECS, as previously reported in the bicarbonate transport

metabolon [7, 8, 12].” – please correct this sentence. Is the suggested change ok?

Lines 302-303: “These results suggest that the interactions of CAII, CAIV and NBC accelerate the rate of HCO3- influx in c-hNECs. “ CAII is cytoplasmic and CAIV membrane-bound but extracellular. Thus CAII cannot interact with CAIV. Perhaps: “interactions of CAII and CAIV with NBC”?

Discussion

Line 364: spelling: small transaient increase

Line 392: „c-hNECs. The Na+/H+ exchange (NHE) is unlikely to extrude H+ from c-hNECs, because it has been shown to be inactive at pHi higher than 7.4 [20].” Yes, but the Authors showed that under zero-CO2 conditions, pHi drops to 7.24 if I am correct. Can NHE be active temporary, depending on the pHi?

Comments on the Quality of English Language

Some parts of the manuscript are still difficult to follow. I suggested a number of corrections to improve it

Author Response

ijms-3100747- corrected version

A decrease in ciliary motility induced by a CO2/HCO3--free solution characteristic of ciliated human nasal epithelial cells: acceleration of HCO3- transport by carbonic anhydrase IV

The clarity of the manuscript was partly improved however, I still found some problems (see below).  Some parts of the discussion can be transferred to the introduction to provide a better background for the reader. 

Major concern.

  1. The Authors stated that “The goal of this study is to clarify the CAIV-mediated mechanism, which suppresses increases in CBF, CBD and pHi in c-hNECs upon applying the Zero-CO2. Is it possible that other membrane-bound CAs (CAIX) could take over CAIV functions and thus the relation between the observed phenotype and CAIV is not that straightforward. CA IX was detected in the cytoplasm and plasma membrane of epithelial cells (Kim et al., 2008).

In the preliminary experiments, we examined mRNAs of 10 CAs (I, II, III, IV, Vb, VI, VII, IX, XII, XIV) by RT-PCR. The CAIX mRNA expressed in both c-hNECs and c-hBECs. The interactions of CAIX with AE2 have been reported in cotransfected HEK239 cells with AE2 or SLC26A7. However, the interactions of CAIX with NBC still remain uncertain. In c-hBECs, the HCO3- transport was not enhanced. Based on these findings, CAIX is unlikely to contribute the enhancement of the HCO3- entry in c-hNECs

We added results of the preliminary experiments (RT-PCR examination of 10 CAs) as supplementary figure (S1). We replaced the previous Table 1 with the new Table 1 showing the PCR primers of 10 CAs.

I changed the results (2.1 2.1. CAIV expression in c-hNECs but not in c-hBEcs) as follows.

“Eleven CA subtypes (I, II, III, IV, Va, Vb, VI, VII, IX, XII and XIV) have been shown to be expressed in normal nasal mucosa [6]. In the preliminary experiments, we examined expressions of ten CA isozyme mRNAs except CAVa in c-hNECs and c-hBECs by Reverse Transcription-Polymerase Chain Reaction (RT-PCR) (S1). The primers used were shown in Table 1. The mRNAs of ten CA subtypes were expressed in c-hNECs. Interestingly, in contrast to other analyzed CAs, the mRNA of CAIV was not expressed in c-hBECs. Kim et al. also reported that mRNAs of CA II, IV and Vb were highly expressed in normal nasal mucosa [6]. Based on these findings, the mRNA expressions of five CA isozymes (I, II, III, IV and Vb) were examined by the RT-PCR in c-hNECs (Fig. 1A) and c-hBECs (Fig. 1B). The mRNA of CAIV is expressed in c-hNECs, but not in c-hBECs. The analysis of CAIV mRNA expressions by Real-Time PCR revealed that mRNAs were significantly higher in c-hNECs than in c-hBECs. (Fig. 1C).

The mRNA expression levels of CA isoforms were low in nasal polyp compared with normal nasal mucosa [6]. In this study, we used two cell types of c-hNECs (uncinate process and nasal polyp) of chronic sinusitis (CS) patients. The mRNAs of five CA subtypes were similarly expressed in two cell types of c-hNECs. “

  1. The Authors used two CA inhibitors, brinzolamide (inhibits CA II) and dorzolamide (inhibits CA II and CAIV).  They found that “brinzolamide and dorzolamide similarly inhibit the Eq.1 in c-hNECs….” and hypothesized that therefore “The similar effects of brinzolamide and dorzolamide on CBF, CAIV may interact with CAII in c-hNECS, as previously reported in the bicarbonate transport metabolon [7, 8, 12].” The identical/ similar effect of the inhibition of CAII and CAII + CAIV may also suggest that CII is responsible for the observed alterations, especially that CAIV is expressed at the very low level. The synthesis of the specific CAIV inhibitors was recently reported (Carbonic Anhydrase IV Selective Inhibitors Counteract the Development of Colitis-Associated Visceral Pain in Rats. Lucarini E, Nocentini A, Bonardi A, Chiaramonte N, Parisio C, Micheli L, Toti A, Ferrara V, Carrino D, Pacini A, Romanelli MN, Supuran CT, Ghelardini C, Di Cesare Mannelli L. Cells. 2021 Sep 26;10(10):2540. doi: 10.3390/cells10102540. Is there an inhibitor that specifically inhibits CAIX?

The similar effects of Brinzolamide and Dorzolamide on CBF show the interactions of CAII and CAIV with NBC. I have corrected in the results and the discussion.

We checked the recent report of cells. Unfortunately, it is difficult to synthesize the CAIV inhibitor in our lab. We gave up the synthesis of the CAIV inhibitor.

There are some inhibitors for CAIX. We did not examine the effects of CAIX inhibitor in this study. Because CAIX is expressed in c-hBECs. 

  • It is confusing why the same conditions could cause “decrease or a small transient increase of CBF (discussion, lines 364-365).

For measurements of pHi, CBF and CBD, we keep cells in the control solution without aeration in the room temperature (22-25°C). This condition appears to decrease HCO3- entry via NBC, and gradually to decrease pHi in c-hNECs. We did experiments within 2-3 hrs. The pHi just before the start of experiments is the important factor, A low pHi induces a small transient increase in CBF and a high pHi induces a gradual decrease in CBF upon applying the Zero-CO2. We added the discussion (p. 13 3rd paragraph)

Minor issues:

Abstract: needs to be partly re-written (for clarity reason). Suggestion: start with the periodic exposure of Nasal epithelial cells to low CO2 (breathe in). Then ask a question how such a conditions affect cilia beating or how cells respond to such conditions. Summarize data and follow with the short conclusion.

I rewrote the abstract.                                      

Introduction

Line 42: - missing comma: ….AE), and Na+/H+ exchanger…

Line 43: spelling mistake: the cullualr metabolism, 

I corrected.

Line 42-43: please add that CO2 is transported from the extracellular fluids via diffusion (correct?) and that its accumulation inside the cell, will cause cell acidification (High level of H+, low pHi). - is this correct? 

I removed this sentence, and the first paragraph of the introduction changed as follows;

Intracellular pH (pHi) regulates various cellular functions, including airway ciliary beating [1, 2]. In many cell types including the airway ciliated epithelial cell, the pHi is controlled by CO2 and ion transporters, such as Na+-HCO3-cotransporter (NBC), Cl-/HCO3- exchanger (anion exchanger, AE), and Na+/H+ exchanger (NHE). The pHi is mainly controlled by the reaction mediated by carbonic anhydrase (CA) (Eq. 1).

Line 45: could you add H2O to the equitation?  (H2O + CO2…..

We changed as follows; CO2 + H2O 1  H+ + HCO3-             (Eq. 1)

Lines 47 and 48: “CO2 entered” and „ HCO3- entered” – did Authors mean increase of the level of…CO2 or HCO3-? If yes, could you add how it happens (diffusion, product of metabolism – CO2), membrane exchanger, co-transporters (HCO3-) ?

I changed to follows;

The CA is an enzyme that catalyzes the hydration and dehydration of CO2. In general, experimental procedures, such as the switch to a CO2/HCO3--free solution (Zero-CO2) from a CO2/HCO3- containing solution (control solution) (removal of CO2 in extracellular fluid) and activation of Na+ HCO3- cotransport (NBC) (HCO3- entry) shift the Eq. 1 to the left (increase in pHi), and contrary, a procedure such as the switch to the control solution from the Zero-CO2 shifts the Eq. 1 to the right (decrease in pHi).

Line 50: spelling mistake: “epitheliium”  

Lines 50 and 51: “…The ciliated nasal epitheliium is an unique tissue placed under an unusual CO2 conditions”  - maybe the Authors could add (if the information is correct) that the level of CO2 cycles between 0.04% (breathe in) and 1% (breathe out)

Lines 58-60: “The periodic air exposure (0.04% CO2) appears to increase pHi leading to a CBF increase in the ciliated nasal epithelial cells, as shown in tracheal and lung airway ciliated cells upon applying the CO2/HCO3--free solution (Zero-CO2) [1, 4, 5].” – please provide this information earlier, just after “The ciliated nasal epitheliium is an unique tissue placed under an unusual CO2 condition: the apical surface is periodically exposed to the fresh air containing an extremely low concentration of CO2 (0.04%) with respiration.”

 I changed as follows (p.2. the top of 2nd paragraph)

The ciliated nasal epithelium is a unique tissue exposed to a low temperature air [3] and air with an extremely low CO2 concentration. The CO2 concentration of apical surface is changed from 0.04% (fresh air) to 1% (exhaled air) with non-exercised respiration. The periodic air exposure (0.04% CO2) appears to increase pHi leading to a CBF increase in the ciliated nasal epithelial cells, as shown in ciliated airway epithelial cells upon applying the CO2/HCO3--free solution (Zero-CO2) [1, 4, 5].

Lines 62-63: ” Ciliated human nasal epithelial cells (c-hNECs) were differentiated from human nasal epithelia (operation samples) by the air liquid interface (ALI) culture.” – as it is now this information belongs rather to the Methods section. Suggested change: “Ciliated human nasal epithelial cells (c-hNECs) are differentiated by the air liquid interface (ALI) culture.” 

Line 61-62. I changed to “Ciliated human nasal epithelial cells (c-hNECs) were differentiated by the air liquid interface (ALI) culture.”

Lines 63 and next. Could you please combine information from [REF2] and next clearly highlight what is different in tracheal cilia [REF1]

 Line 62-64  I changed to follows: In c-hNECs, an application of Zero-CO2 induced small transient increases in pHi, CBF and CBD [2], while it induced large sustained increases in pHi and CBF in the ciliated airway epithelial cells [1].

Line 72: “A previous study showed that 11 CA subtypes are ex-…” - please for clarity describe the CAs in a new paragraph. While describing CAs, please add that CAII is cytoplasmic and CAIV is membrane bound. Please shift the following information from the discussion section to Introduction: “The C-terminal tail of CAIV is anchored in the outer surface of the plasma membrane and a physical interaction between extracellular CAIV and NBC1 occurs via fourth extracellular loop of NBC1 [7] and perhaps include a modified fig 12 showing CAII, CAIV, and cotransporters  localization. It will be very helpful for the reader outside the field to understand from the very beginning that CAIV activity changes the levels of CO2, H+, and HCO3- outside the cell, in the nasal mucous layer and these extracellular changes affect intracellular pHi and cilia beating (is this correct?)

We rewrote the introduction as follows. (p.2 Line 74-86)

We added Fig. 12C.

No one measure pH, CBF during respiration. I think CBF and pH are probably maintained at an adequate level without fluctuation during respiration. Therefore, the HCO3- transport metabolon appears to be essential for maintain CBF, CBD and pHi in nasal epithelia.

Lines 67-68: ” Moreover, an application of NH4+ pulse induced gradual decreases in pHi and CBF following to their immediate increases, suggesting that H+ is produced during the NH4+ pulse [2]. – please explain why (for the reader outside the field) – does NH4+ competes with H+ to bind HCO3-? Does in happens inside the cell or in the mucous layer? The answers to these questions are likely obvious to the Authors but might be not to the reader with general cilia knowledge but unfamiliar with pHi regulation.

The procedure of NH4+ pulse is the addition of 25 mM NH4+Cl. NH4+Cl contains very small amount of free NH3. NH3 enters cell and binds H+ to produce NH4+. NH4+ binds HCO3- to produce ammonium bicarbonate (NH4HCO3). In genral, addition of NH4+Cl reaches an equilibrium (a high pH) within a short time. This method is widely used to clamp a high pH. And After the removal of NH4+ pulse is used for an acid load of cells. The results obtained by c-hBECs clearly show pH was sustained without any decrease. The decrease in pHi is induced by increasing H+ concentration. A gradual decrease in pHi means a gradual increase in [H+]i. This is the goal of this study.

Line 73:” However, the role of CAIV is not fully understood in nasal epithelia,…”. The Authors stated in previous sentence that “…including CAII and CAIV”. The reader may not understand why the presence of CAII and CAIV in nasal epithelial cells was highlighted. If these two were not previously studied, why in the following sentence, Authors refer only to CAIV? If CAII was studied, please provide the information, if not say that CAII and CAIV were not investigated but here we focused on CAIV because…… Is the role of other CAs in nasal epithelial cells “fully understood” in contrast to CAIV? Or maybe the Authors selected CAIV as a continuation of the previous study? Claiming such a research continuation also can be a good explanation why these studies were undertaken. 

I have removed CAII from the sentence. (p.2 Line 74-75)

Line 74:” Previous studies also…” – please remove “also”

Line 76:” in HEK293 cells transfected CAIV and HCO3- transporters” – suggestion “in co-transfected HEK293 cells”

I have deleted (p.2 line 74) and changed (p2. Line 75).

Line 77: “Nasal epithelia have also been shown…” – suggestion ”Moreover, it was shown that NBC [6] and AE [11] are expressed in nasal epithelia”

Line 78:” In c-hNECs, CAIV…” – suggestion, “Thus, In c-hNECs, CAIV…”

I have changed (p. 2, Line 85-86).

Results

Line 92:” 2.1. “CAIV expression in c-hNECs and not in c-hBEcs” – suggestion:” CAIV expression in c-hNECs but not in c-hBEcs”

I would also suggest to remove subchapters titles (2.1.1, 2.1.2, 2.1.3) and simply start description of detection by IF or WB as a new paragraph. 

Line 94: suggest to change:  “The mRNA of CAIV…” – suggested change: “Interestingly, in contrast to other analyzed CAs, the mRNA of CAIV….”….The analyses of the expressionlevels of CAIV by Real-Time PCR revealed that mRNA were significantly higher in c-hNECs than in c-hBECs.

I added the results of preliminary experiments. So I rewrote the first paragraph of p.3, as follows.

“Eleven CA subtypes (I, II, III, IV, Va, Vb, VI, VII, IX, XII and XIV) have been shown to be expressed in normal nasal mucosa [6]. In the preliminary experiments, we examined expressions of ten CA isozyme mRNAs except CAVa in c-hNECs and c-hBECs by Reverse Transcription-Polymerase Chain Reaction (RT-PCR) (S1). The primers used were shown in Table 1. The mRNAs of ten CA subtypes were expressed in c-hNECs. Interestingly, in contrast to other analyzed CAs, the mRNA of CAIV was not expressed in c-hBECs. Kim et al. also reported that mRNAs of CA II, IV and Vb were highly expressed in normal nasal mucosa [6]. Based on these findings, the mRNA expressions of five CA isozymes (I, II, III, IV and Vb) were examined by the RT-PCR in c-hNECs (Fig. 1A) and c-hBECs (Fig. 1B). The mRNA of CAIV is expressed in c-hNECs, but not in c-hBECs. The analysis of CAIV mRNA expressions by Real-Time PCR revealed that mRNAs were significantly higher in c-hNECs than in c-hBECs. (Fig. 1C).”

Lines 101-103: the Authors describe level of CAs in samples from patients. This fragment need an introduction. E.g. we were wondering if the level of CAs is altered in patients suffering from chronic sinusitis (why these conditions were selected?). What is different (in mucus layer) between healthy individuals and patients suffering chronic sinusitis?

I added the following sentence (p.3 line 111).

The mRNA expression levels of CA isoforms were low in nasal polyp compared with normal nasal mucosa [6].

Line 110- please remove the subtitle. State: “The western blot analyses are consisted with Real-Time PCR data. We used two anti-CAIV antibodies (MAB2186 and AF2186, R&D System).”

 I added  (p.3 line 122).

Line 127: suggested change” a single band detected was decreased to 40 kDa, but was still larger than 33 kD. Thus, the treatment with PNGaseF did not completely brake the glycosylation sites. Another treatment may be required…”

 I added “Thus”, according to the suggestion.

Line 133:” 2.1.3. Immunofluorescence examination for CAIV – please, remove.

This paragraph can be further shortened.

  1. I rewrote as follows (p.4 line 143-149)

 Immunofluorescence examinations for CAIV were carried out in c-hNECs (Fig. 3A) and c-hBECs (Fig. 3B) using AF2186 (anti-CAIV antibody, Figs. 3A1 and 3B1) and ab179484 (anti-alpha-tubulin (AC-tubulin) antibody, a cilia marker) (Figs. 3A2 and 3B2). The double staining showed that cilia existing in the apical surface are positively stained for CAIV and AC-tubulin in c-hNECs (Fig. 3A3). However, in c-hBECs, no immunofluorescence for CAIV was detected in cilia (Figs. 3B1-3). Thus, CAIV exists in apical cilia of c-hNECs, but not in those of c-hBECs.

For example:” Immunofluorescence examinations for CAIV were carried out in c-hNECs (Fig. 3A) and c-hBECs (Fig. 3B) using AF2186, anti-CAIV antibodies and anti-acetylated tubulin antibodies (a cilia marker). The double-labelling analyses clearly showed that cilia present in the apical surface of the c-hNECs (Fig. 3A2) are positively stained for both CAIV and AC-tubulin while in c-hBECs cells cilia the CAIV was detected (Fig. 3B1).

(information regarding DAPI staining can be only in the Fig description as this is general staining of the nuclei).

See above.

Line 150:” 2.2. Expression of NBC and AE mRNAs detected by RT-PCR in c-hNECs and c-hBECs”

Line 153: spelling “exprtessed”  „thet”

„Previous studies have already shown thet NBC and AE are expressed in hunan nasal epithelia [15, 16, 17] and mice bronchiolar epithelia 154 [18]. 

Suggestion to change:” …that agrees with the previous studies 

in hunan (please correct spelling) nasal epithelia [15, 16, 17] and mice bronchiolar epithelia 154 [18].

Line 155 ” ….NDCBE1 and AE2…” – please insert comma “NDCBE1, and AE2”

 I corrected misspellings.

Line 163: please add introductory sentence. For example. First we characterized ciliary beating in HNEC and hBEC cells (follow with description in lines 163-167). 

Next we analyze how low level of CO2 affect cilia beating. Under (what kind of conditions?) In c-hNECs, CBF was 8.39 ± 2.44 Hz (n = 47) and CBD was 62.2 ± 13.8 μm (n = 20) in 163 c-hNECs.

I added (p. 5, Line 168)

“First, the CBF and CBD, as indices assessing the ciliary beating activities, were measured in c-hNECs and c-hBECs.”

Line 166: suggestion: Thus, there was no…(please add “Thus” as this is a conclusion)

 I added “Thus” (p.5 line 174).

Line line 170: suggestion (need introduction” Next, in order to understand….we analyzed how minimal CO2 concentration typical for the air, affect ciia beating and pHi. 

I added the following sentence (p. 5 line 175-176).

Next, we applied a CO2/HCO3--free solution in c-hNECs and c-hBECs, in order to understand the effects of an extremely low CO2 concentration on CBF and pHi.

Line 176: the Authors wrote : “The pHi of c-hNECs in the control solution was 7.64 ± 0.19 (n = 9).” But in the line 213 is:” 6C). The pHi was 7.55 in c-hNECs perfused with the control solution.”  In both cases the Authors mean the same “control solution” if not, can be these solution re-named to avoid confusion? If this is the same solution, why pHi is different?

The values of pHi measured by the BCECF fluorescence have a large SE. The control solution used in this study is the same.

As shown in the Figures, pH has a large SD.

Line 181:” because of no HCO3- entry keeping H+ supply from other systems” – please clarify – because of the low level of CO2 (0.04%) or HCO3- (but inside the cell – so low production due to low CO2? What about the transport of HCO3- by co-exchanger and co-transporters?

Line 183:” caused by the different condition of cells used for the experiments.” – could you be more precise regarding differences in the conditions?

Lines 199-200:” We applied the NH4+ pulse in c-hNECs, upon applying the Zero-CO2. The switch to 

the Zero-CO2 slightly increased CBF, but the increase in CBF was not significant” – it seems that the same information was repeated in this sentence.

The switch to the Zero-CO2 did not increase in CBF.

Lines 253-254: “In the presence of brinzolamide, the NH4+ pulse immediately increased CBF

anf then gradually increased it without any CBF decrease.” – I do not understand this part “anfthen gradually increased it without any CBF decrease.” Is this an increased followed by a decrease?

Line 256:’ In the presence of brinzolamide, the NH4+ pulse increased CBF without any CBF decrease in c-hNECs” – please correct. I do not understand “the increase in cilia beat frequency without any decrease of cilia beat frequency”. 

I changed as follows (p.8 line 268) 

In the presence of brinzolamide, the NH4+ pulse increased CBF without any decrease in c-hNECs,

Lines 267-269:” The similar effects of brinzolamide and dorzolamide on CBF, may suggests that(???) CAIV 267 may interact with CAII in c-hNECS, as previously reported in the bicarbonate transport metabolon [7, 8, 12].” – please correct this sentence. Is the suggested change ok?

I changed the sentence to (p. 8, line 277-280) “The similar effects of brinzolamide and dorzolamide on CBF indicate that the CAII is also involved in the regulation of pHi and CBF in c-hNECS, as previously reported in the bicarbonate transport metabolon [7, 8, 12].”

Lines 302-303: “These results suggest that the interactions of CAII, CAIV and NBC accelerate the rate of HCO3- influx in c-hNECs. “ CAII is cytoplasmic and CAIV membrane-bound but extracellular. Thus CAII cannot interact with CAIV. Perhaps: “interactions of CAII and CAIV with NBC”?

I changed the sentence to (p. 9, line 313-314) “These results suggest that the interactions of CAII and CAIV with NBC accelerate the rate of HCO3- influx in c-hNECs.”

Discussion

Line 364: spelling: small transaient increase

I have rewritten the Discussion.

Line 392: „c-hNECs. The Na+/H+ exchange (NHE) is unlikely to extrude H+ from c-hNECs, because it has been shown to be inactive at pHi higher than 7.4 [20].” Yes, but the Authors showed that under zero-CO2 conditions, pHi drops to 7.24 if I am correct. Can NHE be active temporary, depending on the pHi?

You are correct. In some experimental conditions, NHE probably extrudes H+ in c-hNECs, especially in an acid load after removing the NH4+ pulse. Unfortunately, we did not check the NHE activity after removing the NH4+ pulse. However, NHE is unlikely to extrude H+ to generate an extreme high pHi in c-hNECs.

  1. I changed to follows; (p. 12, line 396-400)

In c-hNECs, the pHi is high, except some experimental condition, such as the long-time exposure of the Zero-CO2 and the removal of the NH4+ pulse. Under these experimental conditions, the Na+/H+ exchange (NHE) may extrude H+ from c-hNECs. However, the NHE is unlikely to function for increasing pHi to an extremely high level in c-hNECs, because it has been shown to be inactive at pHi higher than 7.4 [20].

Comments on the Quality of English Language

Some parts of the manuscript are still difficult to follow. I suggested a number of corrections to improve it

Submission Date

26 June 2024

Date of this review

30 Jul 2024 18:45:10

Round 3

Reviewer 2 Report (New Reviewer)

Comments and Suggestions for Authors

The manuscript requires some small changes. Below few examples that I found

Lines 51-56

“The ciliated nasal epithelium is a unique tissue placed under a low temperature [3] and also an unusual CO2 condition. The CO2 concentration of apical surface is changed  from 0.04% (fresh air) to 1% (exhaled air) with non-exercised respiration. The periodic air  exposure (0.04% CO2) appears to increase pHi leading to a CBF increase in the ciliated  nasal epithelial cells, as shown in tracheal and lung airway ciliated cells upon applying  the CO2/HCO3--free solution (Zero-CO2) [1, 4, 5]it seems like either something is missing or indicated part should be removed.

Lines 58-59

“For example, to maintain CBF without a  decrease induced by a low temperature, ciliated nasal epithelial cells express ….”,

Proposed change: For example, to maintain CBF when cells are exposed to a low temperature, ciliated nasal epithelial cells express….”

Lines 61-62

“..periodical air exposure…” Proposed change “..periodical exposure to low and high CO2 concentration…”

Line 66

Moreover, an application of NH4+ pulse” Proposed change Respectively, an application of NH4+ pulse” to indicate that as in the previous sentence, also here are describe first changes in nasal cells and next in tracheal cells

Line 92

shift the Eq.1 to the right to induce no increase or a negligibly small increase in CBF” Proposed change “shift the Eq.1 to the right to maintain or cause a negligibly small increase in CBF”

Comments on the Quality of English Language

suggest to ask someone unfamiliar with the manuscript to read it to find some minor mistakes in the text

Author Response

The manuscript requires some small changes. Below few examples that I found

Lines 51-56

“The ciliated nasal epithelium is a unique tissue placed under a low temperature [3] and also an unusual CO2 condition. The CO2 concentration of apical surface is changed  from 0.04% (fresh air) to 1% (exhaled air) with non-exercised respiration. The periodic air  exposure (0.04% CO2) appears to increase pHi leading to a CBF increase in the ciliated  nasal epithelial cells, as shown in tracheal and lung airway ciliated cells upon applying  the CO2/HCO3--free solution (Zero-CO2) [1, 4, 5] – it seems like either something is missing or indicated part should be removed.

I have changed to “The periodic air exposure (0.04% CO2) appears to increase pHi leading to a CBF increase in the ciliated nasal epithelial cells, since the application of the CO2/HCO3--free solution (Zero-CO2) has been shown to increase pHi, ciliary beat frequency (CBF) and ciliary bend distance (CBD, an index of amplitude) in tracheal and lung airway ciliated cells [1, 4, 5].”

Lines 58-59

“For example, to maintain CBF without a  decrease induced by a low temperature, ciliated nasal epithelial cells express ….”, 

Proposed change: For example, to maintain CBF when cells are exposed to a low temperature, ciliated nasal epithelial cells express….”

I have changed according the reviewer’s proposal.

Lines 61-62

“..periodical air exposure…” Proposed change “..periodical exposure to low and high CO2 concentration…”

I have changed according the reviewer’s proposal.

Line 66

” Moreover, an application of NH4+ pulse” Proposed change ” Respectively, an application of NH4+ pulse” to indicate that as in the previous sentence, also here are describe first changes in nasal cells and next in tracheal cells

I have changed to “Respectively, an application of NH4+ pulse induced a gradual decrease in pHi and CBF (acetazolamide (an inhibitor of carbonic anhydrase (CA)) -sensitive) following to an immediate increase in c-hNECs [2], but a sustained increase in tracheal airway ciliary cells [1]. The gradual decreases in CBF following to immediate increases indicate decreases in pHi in c-hNECs upon applications of the Zero-CO2 and the NH4+ pulse.”.

Line 92

“shift the Eq.1 to the right to induce no increase or a negligibly small increase in CBF” Proposed change “shift the Eq.1 to the right to maintain or cause a negligibly small increase in CBF”

 I have changed according the reviewer’s proposal.

This manuscript is a resubmission of an earlier submission. The following is a list of the peer review reports and author responses from that submission.

Round 1

Reviewer 1 Report

Comments and Suggestions for Authors

While the impact of intracellular pH on cilia beat is important to analyse, there are many flaws in this study that need to be addressed before if can be reviewed fully. To only pick a few major points:

1. from the abstract it is not clear what the hypothesis is and why this study is important

2. Introduction is confusing, you argue nasal epithelium is exposed to ambient air, but during exhalation CO2 is high. There are so many different conditions, nasal, bronchial, ALI culture with 5% CO2, etc, its hard to follow the main message. What is your point?

3. Your argument with the conflicting results in introduction is not clear to follow

4. line 82-86: not clear what the message is

5. Please provide quantitative qPCR data for multiple donors. From the presented data you can not conclude that bronchial epithelial cells don't express CAIV as not matched donors.

6. Use of nasal epithelial cells from nasal polyps are questionable, this tends to be inflamed tissue which can affect expression. This point needs to be addressed.

7. WesternBlot results are not acceptable. The band at 45kDa that you are arguing is CAIV, is not conclusive. The blot with blocking peptide has a faint band at this MW. R&D shows this AB with a clear band in lung tissue. Additionally, if the argument is that CAIV is glycosylated (which would also be the case in lung tissue used by R&D?), this needs to clarified by PNGase treatment. Methods need to be clearer, reducing conditions, samples boiled before SDS-PAGE? A positive control needs to be provided. n-numbers of donors?

8. IF results not acceptable. In WB it has been demonstrated that the antibody has a lot of unspecific binding. Therefore, IF is mostly down to unspecific binding. n-numbers of donors?

9. n-number of donors needs to be clarified, biological or technical replicates?

Unfortunately, the above points need to be addressed before the rest of manuscript can be reviewed in more detail.

Comments on the Quality of English Language

Checks on English language are required. Some sentences are incomplete and therefore hard to understand. Abstract needs to be re-written.

Reviewer 2 Report

Comments and Suggestions for Authors

This manuscript provides pivotal insights into the bicarbonate transport metabolon's role in modulating intracellular pH and ciliary beat frequency in ciliated human nasal epithelial cells. It elucidates the interplay between CAIV, NBC, and CAII, presenting a novel paradigm for pHi regulation within the nasal epithelium, with broader implications for our comprehension of respiratory health.

While endorsing the publication post-revisions, it is paramount to address the methodological quandaries surrounding CA IV detection. Given the RT-PCR results, the purported absence of mRNA and the insufficient specificity of the CA IV antibody, further verification via alternative antibodies, mass spectrometry, or knockout models is indispensable.

The inconsistencies observed in Figures 5A and 7C, when juxtaposed with previous literature and the manuscript's own findings, present a challenge. Specifically, Figure 5A indicates a decrease in CBD following CO2/HCO3-free solution treatment, which diverges from the increase reported by Inui et al. 2019 [6].

Similarly, Figure 7B and 7C exhibit conflicting data on the impact of brinzolamide on CBF. These anomalies require thorough investigation and reconciliation to align the study's outcomes with established research and to verify internal consistency within the presented data.

Moreover, the paradoxical increase in CBF in the presence of S0859, as shown in Figure 9B, calls for a more in-depth exploration of HCO3- transport mechanisms when NBC is ostensibly inhibited.

A minor point for clarity: the expansion of the abbreviation 'CBD' should be articulated within the results section.

In conclusion, upon satisfactory resolution of these concerns, the manuscript would stand as a substantial contribution to the field.